# PATH MATTERS: UNVEILING GEOMETRIC IMPLICIT BIAS VIA CURVATURE-AWARE SPARSE VIEW OPTIMIZATION

**Canran Xiao**[1][†][*]**, Liaoyuan Fan**[2][†]**, Yanbin Li**[3]**, Jing Tang**[4]**, Peilai Yu**[5]
[1]Shenzhen Campus of Sun Yat-sen University, [2]University of Hong Kong
[3]Beijing University of Post and Telecommunications, [4]Huazhong University of Science and Technology, [5]Ludwig Maximilian University of Munich
`xiaocr3@mail.sysu.edu.cn`

## ABSTRACT

3D Gaussian Splatting (3DGS) has recently emerged as a powerful approach for novel view synthesis by reconstructing scenes as sets of Gaussian ellipsoids. Despite its success in scenarios with dense input images, 3DGS faces critical challenges in sparse view settings, often resulting in geometric inaccuracies, inconsistencies across views, and degraded rendering quality. In this paper, we uncover and address two key implicit biases of 3DGS reconstruction algorithm in sparse-view : (1) the model has a stronger demand for supervision signal toward regions of high curvature, and (2) the model is sensitive to the smoothness of the trajectory of the input views. To tackle these issues, we propose a novel framework that optimizes camera trajectories to maximize curvature coverage while enforcing smooth motion, and we further enhance the informativeness of data through a synthetic view generation process. Extensive experiments on Mip-NeRF 360, DTU, Blender, Tanks & Temples, and LLFF datasets show that our method substantially outperforms state-of-the-art solutions in sparse-view scenarios, both in rendering quality and geometric fidelity. Beyond these empirical gains, our investigation uncovers the subtle ways in which data representation and trajectory planning interact to shape 3DGS performance, offering deeper theoretical insights into the algorithm's inherent biases.

## 1 INTRODUCTION

In recent years, 3D Gaussian Splatting (3DGS) (Kerbl et al., 2023; Chen & Wang, 2024; Fei et al., 2024) has emerged as a powerful approach for novel view synthesis, leveraging Gaussian ellipsoids to model scenes more flexibly than traditional representations (Zhang et al., 2023). 3DGS has broad application prospects in fields such as autonomous driving, embodied intelligence, and world models (Xiao et al., 2025; Ni et al., 2025; Zhao et al., 2025; Jiang et al., 2024; Wang et al., 2025a;b; Zhang et al., 2024b). The efficiency of 3DGS in approximating complex geometries and lighting scenarios makes it particularly appealing for multi-view rendering tasks in VR and AR (Zhai et al., 2024) as well as film production (Peng et al., 2024a; Zhang et al., 2025c; 2026). However, a common challenge in applying 3DGS in the real world is the sparse-view scenario, where collecting dense viewpoint data is noisy, impractical or impossible (Feng et al., 2022; Lu et al., 2024; Ni et al., 2025). Enhancing the performance of 3DGS with a few input images offers a compelling direction for improving the capabilities of 3DGS.

However, 3DGS in such sparse-view scenarios presents deeper challenges beyond reliance on dense sampling (Cheng et al., 2024). A key issue is the algorithm's sensitivity to input data distribution and quality, which can severely degrade performance when data is limited or unevenly sampled (Yao et al., 2024; Liu et al., 2025). This often manifests in geometric inaccuracies, inconsistencies across views, and errors in spatial and photometric representation—limitations that become especially pro-

---

[*]Corresponding author
[†]These authors contributed equally to this work

nounced with sparse observations. To mitigate these issues, some methods have been proposed that enhance supervision during training. These approaches primarily use multi-view consistency to create pseudo labels for unseen rays by re-projection (Zhang et al., 2022; Pan et al., 2023; Zhang et al., 2024a) or warping (Xu et al., 2022; Chen et al., 2025; Zhang et al., 2025a) 3D points to unseen viewpoints. While these techniques offer improvements, they do not fully address the underlying weaknesses of 3DGS in handling sparse and suboptimal data distributions, particularly the inherent constraints and biases that lie in the algorithm that arise due to the complex interplay between data distribution and scene representation.

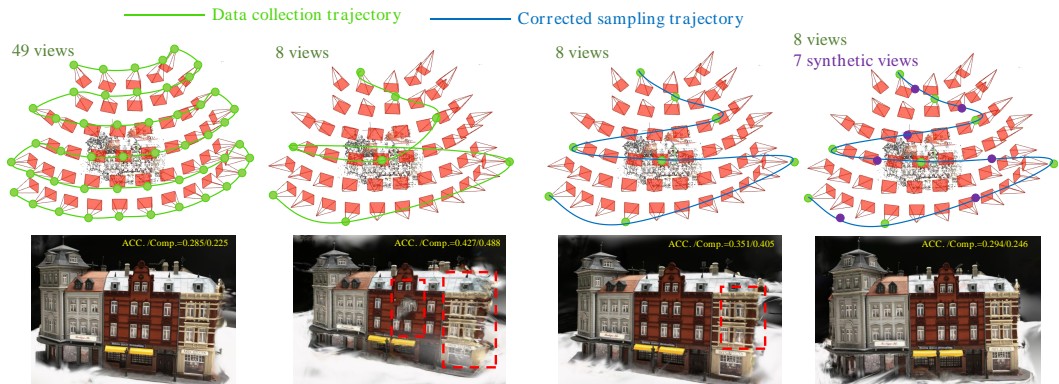

Figure 1: We focus on the problem of 3D reconstruction from sparse views using 3D Gaussian Splatting(3DGS). Our work identifies that sparse view 3DGS is prone to geometric inaccuracies, decreased geometric consistency, and inaccuracies in spatial and photometric properties. This study proposes a framework that enhances the performance of 3DGS in sparse view scenarios, achieving state-of-the-art results.

Although often subtle (Li et al., 2024; Yao et al., 2023), these biases could undermine reconstruction quality when data is limited or unevenly sampled. The interplay between how the scene representations are reconstructed and how viewpoints are arranged remains underexplored. Furthermore, recognizing and mitigating these geometric implicit biases and further reducing these biases to improve the overall reconstruction performance by adapting the learning process is a promising direction in sparse-view reconstruction.

We tackle the above challenges by introducing a novel framework that significantly boosts 3DGS performance in sparse data environments by optimizing the camera trajectory of pseudo labels. Specifically, we reveal how factors like the coverage of scene surface from the synthetic viewpoints and the smoothness of synthetic viewpoints sampling trajectory collectively affect the reconstruction quality under limited observations, and further design strategies, including an optimized path planning method and a synthetic view generation procedure, to mitigate their effects. Beyond simply improving empirical results, our work provides a theoretical understanding of how our method reduces the subtle biases underlying 3DGS.

The key contributions of this paper are as follows: **(i)** We are the first to identify and analyze the inductive biases inherent in Sparse View 3DGS, revealing how biases related to geometric detail prioritization and trajectory smoothness impact the quality of 3D reconstruction from sparse datasets. **(ii)** We propose an optimized camera trajectory framework that maximizes surface curvature coverage while maintaining smoothness, thereby improving the accuracy and consistency of scene reconstruction in sparse view scenarios. **(iii)** We demonstrate the superiority of our approach through extensive evaluations on benchmark datasets, where our method consistently outperforms existing techniques, achieving significant improvements in rendering quality and geometric fidelity.

## 2 RELATED WORK

**3D Gaussian Splatting.** 3D Gaussian Splatting (3DGS) rasterizes anisotropic 3D Gaussians directly onto a 2D screen via a splatting-centric approach (Zwicker et al., 2002), determining pixel colors through depth sorting and alpha blending instead of expensive ray marching. Despite its

near-instant rendering, 3DGS is sensitive to sampling rates; changes in zoom or viewing distance can introduce distortions, mitigated by low-pass filtering (Yu et al., 2023) or multi-level Gaussians (Yan et al., 2023). Moreover, 3DGS often inflates Gaussians beyond true scene geometry, increasing both redundancy and memory usage. Certain approaches manage the impact of Gaussians on the rendering outcome by assessing their size (Lee et al., 2023) or their detectability across different viewpoints (Fan et al., 2023), leading to the elimination of less impactful Gaussians. Alternative strategies minimize the memory footprint of Gaussian parameters through a quantization approach (Navaneet et al., 2023) or by deducing Gaussian attributes from the attributes of a structured lattice (Morgenstern et al., 2023; Lu et al., 2023). Some other works focus on broader aspects of 3DGS models, such as the uncertainty estimation (Li & Cheung, 2024), active learning (Lee et al., 2022), or the copyright protection (Huang et al., 2024).

**Sparse View Rendering.** Synthesizing novel views from sparse images is a significant challenge, as conventional techniques often require an impractically high number of views. Early strategies (Tucker & Snavely, 2020; Zhou et al., 2018; Srinivasan et al., 2019) utilize multi-plane images (MPIs) for depth-based re-rendering. Recent methods leverage the NeRF framework, broadly categorized into two approaches. The first, exemplified by DietNeRF (Jain et al., 2021), enforces semantic consistency across views. RegNeRF (Niemeyer et al., 2022) and ViP-NeRF (Somraj & Soundararajan, 2023) focus on color, depth consistency, and visibility across views. The second approach integrates depth priors, as seen in SparseNeRF (Wang et al., 2023), which uses pre-trained depth estimation for depth regularization, and DSNeRF (Deng et al., 2022), which relies on SfM-derived camera poses for depth guidance. Additionally, Neo 360 (Irshad et al., 2023) adopts a tri-plane feature representation to enhance 3D feature understanding.

## 3  GEOMETRIC IMPLICIT BIASES IN SPARSE VIEW 3D GAUSSIAN SPLATTING

In training a 3DGS model, the distribution of training cameras would influence the model fitting, with the abrupt trajectories causing uneven sampling and increased reconstruction errors. This underscores the need to investigate how trajectory smoothness affects training outcomes, motivating our exploratory work.

### 3.1  EXPLORATORY EXPERIMENTAL SETUP

In order to uncover potential inductive biases in the 3DGS process, we designed a series of experiments focusing on different sampling strategies and camera trajectory configurations. The experiments aim to evaluate how these factors influence the quality of 3D reconstruction under sparse view conditions. We conducted our experiments using a subset of 9 images selected from a total of 106 available images in the Blender (Mildenhall et al., 2021) dataset's LEGO excavator scene. These images were chosen to capture diverse angles and perspectives of the scene, ensuring that key geometric features were represented despite the sparsity of the views. The selected images were used to train the 3DGS model under various conditions, including different image sequences and trajectory paths.

### 3.2  OBSERVATIONS

The experimental results in Figure 2 show the impact of camera trajectory on Sparse View 3DGS reconstruction. The **red** trajectory, with sharp twists and low smoothness, resulted in high error. The **blue** trajectory, with uneven height variations, led to moderate performance. The **orange** trajectory, with distortions and noise, showed the worst reconstruction. In contrast, the **green** trajectory, with smooth motion, achieved the best new view quality. The analysis of the experimental results presented in Figure 2 indicates the following key observations:

**Geometric Detail Prioritization Bias** Dense sampling in object regions of high curvature would benefit the model by reducing the reconstruction errors in these geometrically complex areas. This is evidenced by the higher reconstruction error for the **red** trajectory than the **blue** trajectory, where the sampling was less focused on these critical regions (Zhang et al., 2025b). The corresponding poor reconstruction of surface details reconstructed by the **red** trajectory further validates this bias. This demonstrate that appropriately dense sampling in high curvature areas significantly improves reconstruction quality.

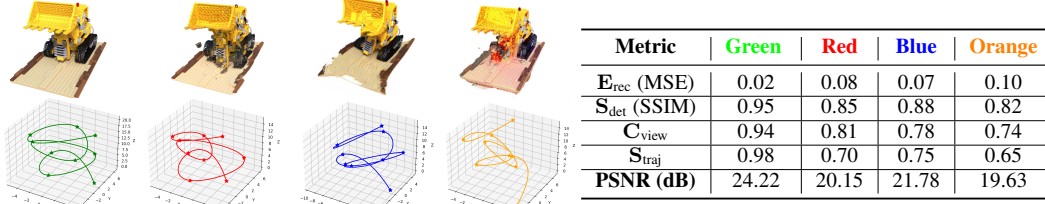

| Metric | Green | Red | Blue | Orange |
|---|---|---|---|---|
| $\mathbf{E}_{\text{rec}}$ (MSE) | 0.02 | 0.08 | 0.07 | 0.10 |
| $\mathbf{S}_{\text{det}}$ (SSIM) | 0.95 | 0.85 | 0.88 | 0.82 |
| $\mathbf{C}_{\text{view}}$ | 0.94 | 0.81 | 0.78 | 0.74 |
| $\mathbf{S}_{\text{traj}}$ | 0.98 | 0.70 | 0.75 | 0.65 |
| PSNR (dB) | 24.22 | 20.15 | 21.78 | 19.63 |

Figure 2: The results of novel view synthesis across different trajectories. The smooth **green** trajectory yields the best overall reconstruction quality. The **red** trajectory exhibits geometric inaccuracies, with loss of details in complex or occluded structures. The **blue** trajectory suffers from decreased geometric consistency across views, leading to poor reconstruction. The **orange** trajectory displays inaccurate reconstruction, highlighting challenges in sparse view reconstruction without bias correction mechanisms.

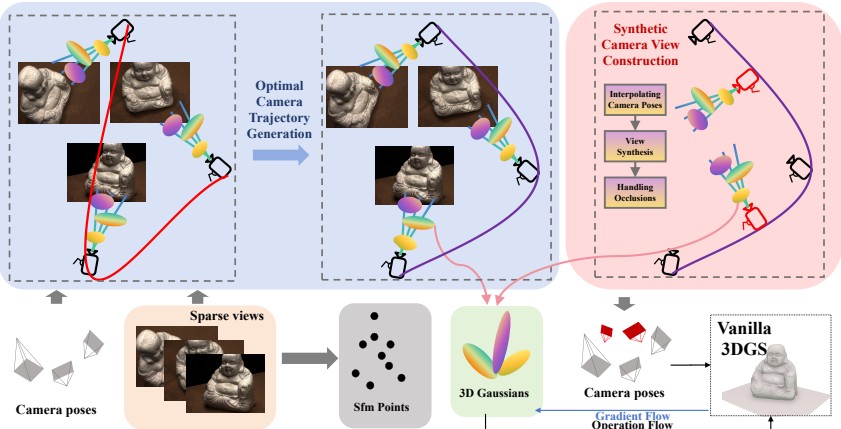

Figure 3: Overview of our Sparse View 3D Gaussian Splatting (3DGS) pipeline. The process starts with sparse camera poses used to generate an optimized camera trajectory, maximizing curvature coverage and ensuring smoothness. Synthetic views are then constructed along this trajectory, enhancing data density. These refined inputs are then processed by the 3DGS framework to reconstruct the final 3D scene.

**Trajectory Smoothness Bias**    Smooth camera trajectories contribute to improved reconstruction stability by minimizing geometric inconsistencies caused by abrupt changes in camera position. This is supported by the fact that minimizing the second derivative of the trajectory would influence the reconstruction quality. The **green** trajectory, with the highest smoothness score, achieved the best overall reconstruction performance. In contrast, the **orange** trajectory, with the lowest smoothness, resulted in the poorest new view quality. These findings confirm that smoother trajectories reduce the impact of noise and irregularities on the reconstruction, leading to higher fidelity.

## 4    A FRAMEWORK FOR SPARSE VIEW 3D GAUSSIAN SPLATTING

In this section, we describe our proposed framework, which addresses the challenges of 3DGS in sparse view settings by optimizing the camera trajectory and constructing synthetic views to enhance scene reconstruction accuracy. Shown in Figure 3, our method consists of two major components: (1) Optimal Camera Trajectory Generation and (2) Synthetic Camera View Construction.

## 4.1 Optimal Camera Trajectory Generation

The goal of our camera trajectory optimization is to maximize the coverage of surface curvature in the scene while ensuring the trajectory's smoothness. Given a sparse sequence of images $\{(V_i, I_i, t_i)\}_{i=1}^{N}$, where $V_i$ represents the camera position of the $i$-th image, $I_i$ is the corresponding image, and $t_i$ is the timestamp, we can construct an initial camera trajectory. This trajectory $\gamma_0(t)$ is defined as the piecewise linear path connecting the camera positions in the sequence.

$$\gamma_0(t) = \sum_{i=1}^{N-1} \mathbf{P}_i \cdot \frac{t_{i+1} - t}{t_{i+1} - t_i} + \mathbf{P}_{i+1} \cdot \frac{t - t_i}{t_{i+1} - t_i} \tag{1}$$

where $\mathbf{P}_i$ represents the camera position at $t_i$. To allow for flexible optimization, the initial trajectory $\gamma_0(t)$ is parameterized using a B-spline curve. This spline is defined by a set of control points $\mathbf{Q} = \{\mathbf{Q}_1, \mathbf{Q}_2, \ldots, \mathbf{Q}_m\}$ and a knot vector $\{t_0, t_1, \ldots, t_m\}$, where the control points are initialized based on the original camera positions $\mathbf{P}_i$.

The B-spline trajectory $\gamma(t)$ is given by:

$$\gamma(t) = \sum_{j=1}^{m} N_j(t) \mathbf{Q}_j \tag{2}$$

where $N_j(t)$ are the B-spline basis functions.

**Curvature and Geometric Feature Calculation**   To optimize the trajectory for capturing important geometric features, we compute the surface curvature at various points on the object's surface. Let $\mathbf{x}$ represent a point on the surface, and $\mathbf{n}(\mathbf{x})$ be the normal vector at that point.

The principal curvatures $\kappa_1(\mathbf{x})$ and $\kappa_2(\mathbf{x})$ at $\mathbf{x}$ are the maximum and minimum curvatures obtained by projecting the surface onto planes that include $\mathbf{n}(\mathbf{x})$. The mean curvature $H(\mathbf{x})$ and Gaussian curvature $K(\mathbf{x})$ are then defined as:

$$H(\mathbf{x}) = \frac{\kappa_1(\mathbf{x}) + \kappa_2(\mathbf{x})}{2} \tag{3}$$

For the trajectory optimization, we focus on mean curvature $H(\mathbf{x})$ as it effectively captures regions with significant geometric features such as edges and corners. We then define the curvature at a point on the trajectory $\gamma(t)$ as: $\kappa(\gamma(t)) = H(\mathbf{x}(\gamma(t)))$, where $\mathbf{x}(\gamma(t))$ is the point on the object's surface corresponding to the trajectory point $\gamma(t)$.

**Optimization Problem Formulation**   The optimization problem is then defined to improve this initial trajectory by maximizing the curvature coverage $\kappa(\gamma(t))$ while adhering to a smoothness constraint. The optimization variables are the positions of the control points $\mathbf{Q}_j$. The optimization problem is formulated as:

$$\underset{\mathbf{Q}}{\text{Maximize}} \quad \int_{t_1}^{t_N} w(\gamma(t)) \|\gamma'(t)\| \, dt \tag{4}$$

Subject to:

$$\int_{t_1}^{t_N} \|\gamma''(t)\|^2 \, dt \leq \epsilon; \qquad\qquad \gamma(t_i) = V_i \quad \text{for } i = 1, 2, \ldots, N$$
$$\|\gamma'(t)\| \geq v_{\min} \quad \text{for all } t \in [t_1, t_N] \quad ; \qquad \kappa(\gamma(t)) \geq \kappa_{\min} \quad \text{for specific regions}$$
$$\int_{t_1}^{t_N} \left( \|\gamma'(t)\| - \|\gamma_0'(t)\| \right)^2 \, dt \leq \delta \tag{5}$$

where $w(\gamma(t)) = \alpha \cdot \kappa(\gamma(t)) + \beta$ is the curvature-weighted function based on the computed mean curvature $\kappa(\gamma(t))$, $\|\gamma'(t)\|$ represents the speed along the trajectory, and $\epsilon$ is a smoothness threshold. The constraint $\gamma(t_i) = V_i$ ensures that the optimized trajectory passes through the original

camera positions. Additionally, $v_{\min}$ enforces a minimum camera speed to prevent the trajectory from stagnating, $\kappa_{\min}$ imposes a minimum curvature constraint in regions of high geometric complexity to enhance sampling, and $\delta$ limits the deviation from the initial trajectory $\gamma_0(t)$, ensuring that the optimized path remains reasonably close to the original.

## 4.2 SYNTHETIC CAMERA VIEW CONSTRUCTION

**Sampling along the optimized path.** Given the optimized trajectory $\gamma^*(t)$, we sample $M$ timestamps $\{t_j\}_{j=1}^{M}$ by arc-length parametrization to ensure approximately uniform baselines. Each synthetic camera pose is

$$V_j^* = \big(\mathbf{P}_j^*,\, \mathbf{R}_j^*,\, \mathbf{K}_j\big), \qquad \mathbf{P}_j^* = \gamma^*(t_j),\ \ \mathbf{R}_j^* = \text{SLERP}\big(\mathbf{R}_a, \mathbf{R}_b;\, \lambda_j\big), \tag{6}$$

where $(\mathbf{R}_a, \mathbf{R}_b)$ are the two neighboring real orientations bracketing $t_j$ and $\lambda_j \in [0,1]$ is the normalized interpolation weight. Unless stated otherwise, $\mathbf{K}_j$ is inherited from the temporally nearest real view.

For warping-based synthesis we require a per-pixel depth map for each real source view $I_i$. We obtain and refine these depths in two stages, please refer to §A.4 for details.

**View Synthesis via Parallax-Based Interpolation** For each real source $(I_i, V_i)$ with refined depth $D_i$, we warp pixels to the synthetic view $V_j^*$. Let $\bar{\mathbf{u}}_i = [u_i, v_i, 1]^\top$ and $\mathbf{K}_i$ be the intrinsics. The 3D point in camera-$i$ coordinates is

$$\mathbf{x}_i(\mathbf{u}_i) = D_i(\mathbf{u}_i)\,\mathbf{K}_i^{-1}\,\bar{\mathbf{u}}_i. \tag{7}$$

Transforming to camera-$j$ and projecting:

$$\mathbf{x}_j = \mathbf{R}_{ij}\,\mathbf{x}_i + \mathbf{t}_{ij}, \quad \tilde{\mathbf{u}}_j = \mathbf{K}_j\,\mathbf{x}_j, \quad \mathbf{u}_j = \Pi(\tilde{\mathbf{u}}_j), \ \ z_{ij}(\mathbf{u}_i) = \big[\mathbf{x}_j\big]_z, \tag{8}$$

where $(\mathbf{R}_{ij}, \mathbf{t}_{ij})$ is the relative pose and $\Pi$ denotes homogeneous normalization. We obtain the color $\tilde{I}_i(\mathbf{u}_j)$ by sampling $I_i$ at the pre-image of $\mathbf{u}_j$ with bicubic interpolation.

The final synthetic image is a depth- and visibility-weighted blend of all warped sources:

$$I_j^*(\mathbf{u}) = \frac{\sum_{i=1}^{N} w_{ij}(\mathbf{u})\,\tilde{I}_i(\mathbf{u})}{\sum_{i=1}^{N} w_{ij}(\mathbf{u})}, \quad w_{ij}(\mathbf{u}) = M_{ij}(\mathbf{u})\,e^{-\lambda_d\,|D_i(\mathbf{u}) - z_{ij}(\mathbf{u})|}\,\max\big(0, \langle \mathbf{v}_i, \mathbf{v}_j \rangle\big), \tag{9}$$

where $\mathbf{v}_i, \mathbf{v}_j$ are view directions and $M_{ij}$ is a z-buffer visibility mask computed with a small tolerance to suppress ghosting. Small holes after blending are filled by edge-aware, depth-guided inpainting.

## 5 EXPERIMENTAL RESULTS

### 5.1 EXPERIMENTAL SETUP

**Dataset.** We evaluate our method on the DTU dataset (Jensen et al., 2014), which contain various materials, textures, and complex geometry shapes. Originally, each scene contains 49 or 64 images. We also reconstruct all scenes from the Blender dataset (Mildenhall et al., 2021) using our method. Each scene in the Blender dataset contains 100 training images, and scenes are generated from a physical-based rendering engine and include effects such as reflections. Moreover, we also evaluate our method on the Mip-NeRF 360 dataset (Barron et al., 2022), which is composed of nine distinct scenes, with five set outdoors and four indoors. We also evaluate our method on the LLFF (Mildenhall et al., 2019) and Tanks & Temples (Knapitsch et al., 2017) dataset.

**Implementation Details.** Our implementation extends the 3D Gaussian Splatting (3DGS) framework (Kerbl et al., 2023) with enhancements tailored for sparse view challenges. We parameterize the initial camera trajectory with B-spline curves, optimized via L-BFGS (Berahas et al., 2016) to maximize surface curvature coverage and maintain smoothness while adhering to the original path. Gaussian parameters $\{\mathbf{p}, \mathbf{c}, \alpha\}$ were estimated directly from sparse views. Optimization used the Adam optimizer with a learning rate of $1 \times 10^{-4}$, training for 150k iterations with a batch size of 2048 on NVIDIA A100 GPUs.

Table 1: Quantitative comparison on **Mip-NeRF 360** (12 views) and **Tanks & Temples** (3 views).

| Model | Venue | Mip-NeRF 360 (12 views) | | | Tanks & Temples (3 views) | | |
|---|---|---|---|---|---|---|---|
| | | PSNR ↑ | SSIM ↑ | LPIPS ↓ | PSNR ↑ | SSIM ↑ | LPIPS ↓ |
| NeuS (Wang et al., 2021) | NeurIPS'21 | 18.75 | 0.50 | 0.47 | 20.10 | 0.64 | 0.34 |
| NeuralAngelo (Li et al., 2023) | CVPR'23 | 18.52 | 0.48 | 0.47 | 20.50 | 0.66 | 0.33 |
| 3DGS (Kerbl et al., 2023) | SIGGRAPH'23 | 15.14 | 0.40 | 0.56 | 17.90 | 0.55 | 0.40 |
| SparseGS (Xiong et al., 2023) | arXiv'23 | 15.93 | 0.42 | 0.46 | 22.00 | 0.72 | 0.30 |
| DNGaussian (Li et al., 2024) | CVPR'24 | 15.66 | 0.41 | 0.48 | 20.80 | 0.67 | 0.33 |
| MatSparse3D (Mao et al., 2024) | CVPR'24 | 15.95 | 0.41 | 0.47 | 21.30 | 0.69 | 0.31 |
| InstantSplat (Fan et al., 2024) | arXiv'24 | 17.90 | 0.46 | 0.48 | 22.10 | 0.73 | 0.29 |
| SuGar (Guédon & Lepetit, 2024) | CVPR'24 | 19.02 | 0.51 | 0.46 | 22.70 | 0.75 | 0.27 |
| VCR-GauS (Chen et al., 2024b) | NeurIPS'24 | 19.21 | 0.51 | 0.45 | 22.90 | 0.76 | 0.25 |
| PGSR (Chen et al., 2024a) | TVCG'24 | 19.17 | 0.51 | 0.45 | 23.00 | 0.76 | 0.25 |
| SCGaussian (Peng et al., 2024b) | NeurIPS'24 | 19.70 | 0.55 | 0.43 | 22.17 | 0.75 | 0.26 |
| MVPGS (Xu et al., 2024) | ECCV'24 | 19.85 | 0.55 | 0.44 | 25.57 | 0.85 | 0.14 |
| NexusGS (Zheng et al., 2025) | CVPR'25 | 19.40 | 0.54 | 0.43 | 25.20 | 0.84 | 0.15 |
| **3DGS + Ours** | – | 19.95 | **0.56** | 0.42 | **26.41** | **0.86** | 0.14 |
| **SCGaussian + Ours** | – | 20.05 | **0.56** | **0.41** | 23.02 | 0.77 | 0.24 |
| **MVPGS + Ours** | – | **20.15** | **0.56** | 0.42 | 26.10 | **0.86** | **0.13** |

Table 2: Quantitative comparison on **DTU** with different training ratios $\alpha$ (Used/All). Lower is better for CD and LPIPS. LPIPS is computed with the standard VGG-based LPIPS .

| Model | $\alpha = 0.2$ | | | $\alpha = 0.3$ | | | $\alpha = 0.4$ | | |
|---|---|---|---|---|---|---|---|---|---|
| | PSNR ↑ | CD ↓ | LPIPS ↓ | PSNR ↑ | CD ↓ | LPIPS ↓ | PSNR ↑ | CD ↓ | LPIPS ↓ |
| NeuS (Wang et al., 2021) | 23.72 | 4.50 | 0.31 | 24.55 | 3.98 | 0.28 | 25.49 | 3.67 | 0.26 |
| NeuralAngelo (Li et al., 2023) | 24.10 | 4.30 | 0.30 | 25.12 | 3.80 | 0.27 | 26.20 | 3.50 | 0.25 |
| 3DGS (Kerbl et al., 2023) | 22.14 | 5.64 | 0.37 | 23.31 | 3.39 | 0.33 | 24.64 | 3.29 | 0.31 |
| SparseGS (Xiong et al., 2023) | 25.23 | 4.42 | 0.26 | 26.19 | 3.45 | 0.23 | 26.56 | 3.71 | 0.24 |
| DNGaussian (Li et al., 2024) | 24.61 | 4.12 | 0.28 | 24.89 | 4.36 | 0.29 | 25.33 | 3.99 | 0.27 |
| MatSparse3D (Mao et al., 2024) | 25.75 | 5.49 | 0.27 | 25.86 | 4.64 | 0.25 | 25.99 | 4.23 | 0.24 |
| InstantSplat (Fan et al., 2024) | 23.10 | 4.12 | 0.32 | 24.23 | 3.90 | 0.29 | 25.37 | 3.85 | 0.27 |
| SuGar (Guédon & Lepetit, 2024) | 24.80 | 4.20 | 0.27 | 25.48 | 3.60 | 0.24 | 26.55 | 3.44 | 0.23 |
| VCR-GauS (Chen et al., 2024b) | 25.02 | 4.18 | 0.26 | 25.90 | 3.50 | 0.24 | 26.70 | 3.40 | 0.22 |
| PGSR (Chen et al., 2024a) | 25.05 | 4.16 | 0.26 | 25.85 | 3.48 | 0.24 | 26.64 | 3.42 | 0.22 |
| NexusGS (Zheng et al., 2025) | 26.10 | 3.65 | 0.22 | 26.65 | 3.28 | 0.21 | 27.10 | 3.18 | 0.20 |
| **Ours** | **27.05** | **3.49** | **0.21** | **27.38** | **3.07** | **0.19** | **27.89** | **3.01** | **0.18** |

**Number of Images.** This paper analyzes how reconstruction quality changes with varying image counts, particularly in low-image scenarios. Unlike prior studies that used a fixed number of images per scene, we employ a proportional approach, introducing the ratio $\alpha$ to represent the fraction of images used relative to the total dataset. The dataset is split, with 70% for training and 30% for testing. Subsets with $\alpha$ set at 20%, 30%, and 40% are used for image rendering and surface reconstruction assessments. For the Mip-NeRF 360 dataset, we utilize 12 views for each scene.

## 5.2 EXPERIMENTAL RESULTS

### 5.2.1 QUANTITATIVE PERFORMANCE

As shown in Tables 1 and 2, and further validated by our Blender dataset results, our method consistently outperforms state-of-the-art approaches across datasets and training ratios. On DTU with $\alpha = 0.4$, it achieves the highest PSNR (27.89) and lowest Chamfer Distance (CD) (3.01), demonstrating superior image quality and mesh accuracy. Even with fewer training images, it remains competitive, outperforming SparseGS, Dngaussian, and MatSparse3D, especially on complex geometries. On Mip-NeRF 360, our plug-in variants achieve the best PSNR of 20.15 and the lowest LPIPS of 0.41 among the compared methods, demonstrating high-fidelity synthesis with strong perceptual quality. Furthermore, we demonstrate the model-agnostic nature of our approach by integrating it with existing frameworks. As detailed in Table 1, combining our method with MVPGS and SC-Gaussian (denoted as MVPGS + Ours and SCGaussian + Ours) yields consistent performance gains (e.g., boosting SCGaussian by 0.85 dB on Tanks & Temples), verifying that our curvature-aware optimization and synthetic densification strategy can serve as a plug-and-play module to enhance various Gaussian splatting backbones.

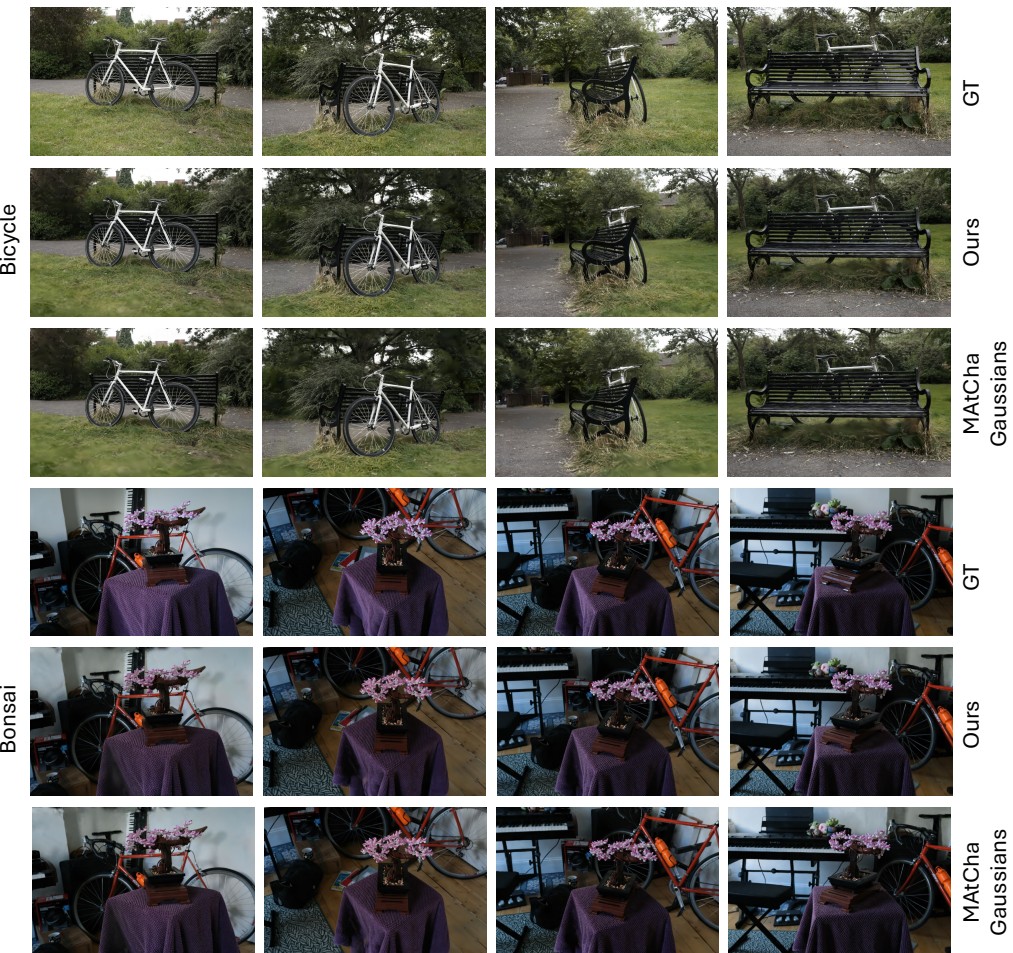

Figure 4: The visualization results on the MipNeRF 360 dataset. We visualize the reconstruction results on testing views with 12 total views.

Table 3: Three-view comparison on **DTU** and **LLFF**.

| Method | Venue | DTU (3 views) | | | LLFF (3 views) | | |
|---|---|---|---|---|---|---|---|
| | | PSNR ↑ | SSIM ↑ | LPIPS ↓ | PSNR ↑ | SSIM ↑ | LPIPS ↓ |
| SparseNeRF (Wang et al., 2023) | CVPR'23 | 19.55 | 0.77 | 0.20 | 19.86 | 0.62 | 0.33 |
| SPARF (Truong et al., 2023) | CVPR'23 | 17.75 | 0.68 | 0.33 | 20.20 | 0.63 | 0.33 |
| MVPGS (Xu et al., 2024) | ECCV'24 | 20.54 | 0.73 | 0.19 | 20.65 | 0.88 | 0.10 |
| SCGaussian (Peng et al., 2024b) | NeurIPS'24 | 20.56 | 0.86 | 0.12 | 20.77 | 0.71 | 0.22 |
| NexusGS (Zheng et al., 2025) | CVPR'25 | 20.40 | 0.88 | 0.26 | 20.80 | 0.77 | 0.24 |
| MAtCha Gaussians (Guédon et al., 2025) | CVPR'25 | 20.52 | 0.86 | 0.27 | 20.74 | 0.75 | 0.24 |
| DIFIX3D+ (Wu et al., 2025) | CVPR'25 | 20.60 | 0.87 | 0.26 | 20.91 | 0.76 | 0.25 |
| **Ours** | – | **20.65** | **0.89** | 0.24 | **20.93** | **0.78** | 0.23 |

We further conduct experiments with only 3 input views on the DTU and LLFF datasets, comparing our method with recent advanced approaches. As shown in Table 3, our method achieves notable performance on both datasets with extremely sparse viewpoints. Specifically, for DTU (3 views), our method attains the highest PSNR (20.65) and SSIM (0.891) among all baselines, demonstrating robustness in reconstructing detailed geometry from limited inputs. On the LLFF dataset (3 views), our method also achieves the highest PSNR of 20.93.

### 5.2.2 Qualitative Performance

Fig. 4 shows qualitative comparisons on the real-world Mip-NeRF 360 scenes *Bicycle* and *Bonsai*. Across all viewpoints, our method produces renderings that are perceptually closer to the ground truth than MAtCha Gaussians (Guédon et al., 2025). In the *Bicycle* scene, our reconstruction better preserves thin structures such as the wheel spokes and the slats of the bench, while maintaining a sharper grass–gravel boundary and more faithful foliage textures. MAtCha Gaussians exhibits noticeable blurring and mild color shifts in these high-frequency regions. In the *Bonsai* scene, our method more accurately reproduces the fine floral details and the textured tablecloth, and matches the global exposure of the room and background objects, whereas MAtCha Gaussians tends to underexpose the scene and oversmooth small structures. These visual results indicate that our approach yields more accurate geometry and appearance reconstruction on complex, large-baseline Mip-NeRF 360 scenes, especially for high-frequency details and challenging real-world backgrounds. More qualitative results can refer to appendix D.

### 5.2.3 Ablation Study

We conducted an ablation study on DTU to assess each component's impact (Table 4a). Removing key elements significantly degrades 3D reconstruction under sparse views. Omitting Optimal Camera Trajectory Generation drops PSNR from 27.05 to 24.20 (20% training ratio) and increases CD, highlighting the need for curvature coverage and smoothness. Excluding Synthetic View Construction further reduces PSNR and raises CD, especially at higher training ratios. Removing the Trajectory Smoothness Constraint lowers PSNR to 26.84. Most critically, without Occlusion Handling, PSNR drops by 2.5 and CD increases by 0.5, underscoring its importance. Visual results are in the supplementary material.

### 5.2.4 Efficiency Analysis

We evaluate efficiency on indoor/outdoor scenes (Table 4b). Compared to MVS-initialized 3DGS, our method improves rendering quality with minimal training overhead. Outdoors, our method improves over the SfM-initialized 3DGS baseline by +3.11 dB PSNR while preserving real-time rendering speed. Indoors, our compact distribution captures details while reaching 35.44 PSNR in 75 minutes, substantially faster than MVS-initialized 3DGS while maintaining comparable real-time FPS. This balances fidelity and efficiency.

Table 4: Results of Ablation Experiments and Efficiency Analysis

(a) An ablation study on the DTU dataset shows the performance of our framework with various components removed

| Ratio (Used/All) | $\alpha = 0.2$ | | $\alpha = 0.3$ | | $\alpha = 0.4$ | |
|---|---|---|---|---|---|---|
| Metrics | PSNR ↑ | CD ↓ | PSNR ↑ | CD ↓ | PSNR ↑ | CD ↓ |
| w/o Optimal Camera Trajectory Generation | 24.20 | 5.12 | 24.58 | 3.95 | 25.32 | 3.62 |
| w/o Synthetic Camera View Construction | 24.38 | 4.98 | 25.01 | 3.72 | 25.90 | 3.45 |
| w/o Trajectory Smoothness Constraint | 25.84 | 3.98 | 26.22 | 3.51 | 26.84 | 3.56 |
| w/o Occlusion Handling | 24.58 | 5.10 | 24.98 | 3.85 | 25.46 | 3.63 |
| Ours | **27.05** | **3.49** | **27.38** | **3.07** | **27.89** | **3.01** |

(b) Efficiency Analysis

| Scene | Strategy | PSNR | Gaussians | Training Time | FPS |
|---|---|---|---|---|---|
| Outdoor | 3DGS (SfM Init) | 34.12 | 640k | 35 min | 115 |
| | 3DGS (MVS Init) | 35.74 | 1612k | 245 min | 70 |
| | Ours (SfM Init) | 37.23 | 980k | 50 min | 110 |
| Indoor | 3DGS (SfM Init) | 32.85 | 1505k | 55 min | 100 |
| | 3DGS (MVS Init) | 33.65 | 1775k | 265 min | 85 |
| | Ours (SfM Init) | 35.44 | 1402k | 75 min | 108 |

## 6 Conclusion

This paper is the first to reveal inductive biases in Sparse View 3D Gaussian Splatting (3DGS), identifying key biases in geometric detail prioritization and trajectory smoothness that affect 3D reconstruction from sparse data. We propose a novel framework integrating optimal camera trajectory generation and synthetic camera view construction, improving accuracy by maximizing curvature coverage and ensuring smooth paths. Experiments on five datasets show significant gains.

**Ethics Statement** This work adheres to the ICLR Code of Ethics. Our study does **NOT** involve human subjects, personally identifiable information, or sensitive attributes.

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

# A  SUPPLEMENTARY TECHNICAL DETAILS

## A.1  DEFINITION OF THE METRICS IN TABLE 1

To systematically evaluate the impact of different configurations, we used several key metrics: reconstruction error $E_{\text{rec}}$ (measured by mean squared error), surface detail preservation $S_{\text{det}}$ (using Structural Similarity Index, SSIM), view consistency $C_{\text{view}}$ (quantifying alignment across perspectives), and trajectory smoothness $S_{\text{traj}}$ (assessed by the second derivative of the camera path).

**Reconstruction Error $E_{\text{rec}}$:**  Reconstruction error quantifies the difference between the predicted and ground truth positions of key points in the 3D model. It is typically measured using Mean Squared Error (MSE). The formula is defined as:

$$E_{\text{rec}} = \frac{1}{N} \sum_{i=1}^{N} \|\mathbf{p}_i^{\text{pred}} - \mathbf{p}_i^{\text{gt}}\|^2 \tag{10}$$

where $\mathbf{p}_i^{\text{pred}}$ is the predicted position of the $i$-th key point, $\mathbf{p}_i^{\text{gt}}$ is the ground truth position, and $N$ is the total number of key points.

**Surface Detail Preservation $S_{\text{det}}$:**  Surface detail preservation assesses the sharpness and fidelity of reconstructed edges and corners, using the Structural Similarity Index (SSIM). The SSIM index is computed as:

$$S_{\text{det}} = \frac{(2\mu_x\mu_y + c_1)(2\sigma_{xy} + c_2)}{(\mu_x^2 + \mu_y^2 + c_1)(\sigma_x^2 + \sigma_y^2 + c_2)} \tag{11}$$

where $\mu_x$ and $\mu_y$ are the mean intensities, $\sigma_x^2$ and $\sigma_y^2$ are the variances, and $\sigma_{xy}$ is the covariance of the pixel values in the predicted and ground truth images. $c_1$ and $c_2$ are constants to stabilize the division.

**View Consistency $C_{\text{view}}$:**  View consistency measures the alignment and coherence of reconstructed surfaces when viewed from different angles. It can be quantified by comparing the depth maps or point clouds across multiple views:

$$C_{\text{view}} = \frac{1}{V} \sum_{i=1}^{V} \frac{1}{M} \sum_{j=1}^{M} \|\mathbf{d}_{ij}^{\text{pred}} - \mathbf{d}_{ij}^{\text{gt}}\|^2 \tag{12}$$

where $V$ is the number of views, $M$ is the number of depth points in each view, $\mathbf{d}_{ij}^{\text{pred}}$ is the predicted depth of the $j$-th point in the $i$-th view, and $\mathbf{d}_{ij}^{\text{gt}}$ is the corresponding ground truth depth.

**Trajectory Smoothness $S_{\text{traj}}$:**  Trajectory smoothness ensures that the camera path does not exhibit abrupt changes in direction. It is assessed by evaluating the second derivative of the camera trajectory:

$$S_{\text{traj}} = \int_{t_1}^{t_N} \|\gamma''(t)\|^2 \, dt \tag{13}$$

where $\gamma(t)$ is the camera trajectory as a function of time $t$, and $\gamma''(t)$ is the second derivative of the trajectory with respect to time, which represents the acceleration or curvature of the path.

These metrics comprehensively evaluate the performance of 3D reconstruction by considering both geometric accuracy and photometric consistency across views.

## A.2  DETAILED SETTINGS FOR THE CURVATURE-COVERAGE STUDY

**Trajectory generation**  The starting trajectory $\gamma^\star \colon [0, 1] \to \mathbb{R}^3$ is a cubic B-spline

$$\gamma^\star(t) = \sum_{j=1}^{M} N_{j,3}(t)\, \mathbf{Q}_j, \quad M = 24, \tag{14}$$

Table 5: Full per-scene metrics at three curvature-coverage levels. Higher PSNR/SSIM and lower Chamfer-D/P2S indicate better quality.

| Scene | PSNR↑ (dB) | | | SSIM↑ | | | Chamfer-D↓ ×$10^{-3}$ | | | P2S↓ | | |
|---|---|---|---|---|---|---|---|---|---|---|---|---|
| | 54.8% | 73.6% | 88.9% | 54.8% | 73.6% | 88.9% | 54.8% | 73.6% | 88.9% | 54.8% | 73.6% | 88.9% |
| Garden | 22.74 | 25.06 | 26.13 | 0.522 | 0.566 | 0.579 | 5.12 | 3.93 | 3.28 | 1.81 | 1.48 | 1.32 |
| Kitchen | 23.13 | 25.55 | 27.26 | 0.534 | 0.589 | 0.620 | 4.55 | 3.62 | 3.14 | 1.72 | 1.39 | 1.26 |
| Stump | 23.56 | 25.78 | 27.46 | 0.550 | 0.594 | 0.617 | 4.86 | 3.64 | 3.37 | 1.78 | 1.46 | 1.34 |
| Bicycle | 23.29 | 25.24 | 27.36 | 0.559 | 0.579 | 0.615 | 5.09 | 3.93 | 3.17 | 1.76 | 1.44 | 1.25 |

where $\{N_{j,3}\}$ are uniform B-spline basis functions of degree 3 and $\mathbf{Q}_j \in \mathbb{R}^3$ the control points optimised by our full method (§4, Eq. (5)). The curve is anchored at the $N_{\mathrm{cam}} = 12$ sparse input poses $\{\mathbf{V}_i\}$ via soft constraints $\sum_i \|\gamma(t_i) - \mathbf{V}_i\|^2$.

1) Step 1 — sub-sampled control-point sets. For level $k \in \{1, 2, 3, 4\}$ we form an index mask $\mathcal{J}_k = \{\, j \mid j \bmod 2^k = 0 \,\}$, i.e. we retain every $2^k$-th control point of $\gamma^\star$ and delete the rest. Let $\mathbf{Q}^{(k)} = \{\mathbf{Q}_j\}_{j \in \mathcal{J}_k}$, $M_k = |\mathcal{J}_k| = 24/2^k$.

2) Step 2 — re-optimisation with fixed smoothness. We re-optimise $\mathbf{Q}^{(k)}$ by solving

$$\min_{\mathbf{Q}^{(k)}} \underbrace{\int_0^1 w\big(\kappa(\gamma_k(t))\big) \|\gamma_k'(t)\| \, dt}_{\text{coverage term}} + \lambda_{\mathrm{curv}} \underbrace{\int_0^1 \|\gamma_k''(t)\|^2 \, dt}_{\text{smoothness}} + \lambda_{\mathrm{anchor}} \sum_{i=1}^{N_{\mathrm{cam}}} \big\|\gamma_k(t_i) - \mathbf{V}_i\big\|^2, \qquad (15)$$

where $\gamma_k$ denotes the spline with control points $\mathbf{Q}^{(k)}$ and the same knot vector as $\gamma^\star$. Crucially, the smoothness weight is *kept identical* ($\lambda_{\mathrm{curv}} = 10^{-2}$) for all $k$, so that the only DoF influencing curvature coverage is the number of control points $M_k$.

3) Step 3 — measuring curvature coverage. Let $\mathcal{S}_{\mathrm{high}}$ be the set of high-curvature triangles described in App. A.2, and $\mathcal{S}_{\mathrm{cov}}(\gamma_k) = \big\{\, \Delta \in \mathcal{S}_{\mathrm{high}} \mid \exists t \colon \mathrm{frustum}(\gamma_k(t)) \cap \Delta \neq \varnothing \,\big\}$. Coverage is then

$$C_k = \frac{|\mathcal{S}_{\mathrm{cov}}(\gamma_k)|}{|\mathcal{S}_{\mathrm{high}}|} \times 100\%. \qquad (16)$$

The above procedure yields the five monotone coverage levels: $C = \{88.9, 84.7, 73.6, 64.1, 54.8\}\%$ for $k = 0 \ldots 4$, respectively.

**Curvature computation** After training with $\gamma^\star$ we extract a 1.5 M-face mesh via marching cubes. Mean curvature $H(\mathbf{x})$ is computed at each vertex using the cotangent Laplacian; triangles whose average $H$ exceeds the 95th percentile are marked "high-curvature". Coverage $C$ is the percentage of these marked triangles intersected by at least one camera frustum.

**Training hyper-parameters** For every $\gamma_k$ we keep: Adam optimiser, learning rate $1 \times 10^{-4}$, batch size 2048, 150k iterations, Gaussian init identical to the main method. Training one model on an NVIDIA A100 takes $\sim$80 min; the entire sweep (5 trajectories × 4 scenes) required 2 GPU-days.

**Expanded quantitative tables** Tables 5 list full image and geometry metrics per scene and per coverage level.

**Correlation analysis** Pearson coefficients averaged over the four scenes are $r_{\mathrm{PSNR}} = 0.96$, $r_{\mathrm{SSIM}} = 0.94$, $r_{\mathrm{CD}} = -0.93$, confirming the near-linear dependency between $C$ and reconstruction quality.

### A.3 PSEUDOCODE

The pseudocode of our method In our framework, the Sparse View 3D Gaussian Splatting (3DGS) optimization process is detailed in Algorithm 1, while the GPU software rasterization of 3D Gaussians is outlined in Algorithm 2.

The goal of the optimization is to adjust the control points $\mathbf{Q}_j$ of the B-spline curve $\gamma(t)$ to maximize the geometric detail captured by the trajectory while maintaining smoothness. The trajectory is parameterized as:

$$\gamma(t) = \sum_{j=1}^{m} N_j(t)\mathbf{Q}_j \tag{17}$$

where $N_j(t)$ are the B-spline basis functions, and $\mathbf{Q}_j$ are the control points.

The objective function to maximize the coverage of geometric features is defined as:

$$J(\gamma) = \int_{t_1}^{t_N} w(\gamma(t))\|\gamma'(t)\| \, dt \tag{18}$$

where $w(\gamma(t)) = \alpha \cdot \kappa(\gamma(t)) + \beta$, $\kappa(\gamma(t))$ is the mean curvature at point $\gamma(t)$, and $\|\gamma'(t)\|$ represents the speed along the trajectory.

The optimization problem is thus:

$$\mathbf{Q}^* = \arg\max_{\mathbf{Q}} \{J(\gamma)\} \tag{19}$$

The optimization employs the constrained L-BFGS method, which requires the computation of the gradient of the objective function with respect to the control points $\mathbf{Q}_j$.

The gradient of the objective function $J(\gamma)$ with respect to $\mathbf{Q}_j$ is given by:

$$\frac{\partial J(\gamma)}{\partial \mathbf{Q}_j} = \int_{t_1}^{t_N} \left( \frac{\partial w(\gamma(t))}{\partial \mathbf{Q}_j}\|\gamma'(t)\| + w(\gamma(t))\frac{\partial\|\gamma'(t)\|}{\partial \mathbf{Q}_j} \right) dt \tag{20}$$

where:

$$\frac{\partial w(\gamma(t))}{\partial \mathbf{Q}_j} = \alpha\frac{\partial \kappa(\gamma(t))}{\partial \mathbf{Q}_j} \tag{21}$$

and:

$$\frac{\partial\|\gamma'(t)\|}{\partial \mathbf{Q}_j} = \frac{\gamma'(t) \cdot \frac{\partial\gamma'(t)}{\partial \mathbf{Q}_j}}{\|\gamma'(t)\|} \tag{22}$$

The curvature $\kappa(\gamma(t))$ at a point $\gamma(t)$ can be expressed in terms of the first and second derivatives of $\gamma(t)$:

$$\kappa(\gamma(t)) = \frac{\|\gamma'(t) \times \gamma''(t)\|}{\|\gamma'(t)\|^3} \tag{23}$$

Thus, the gradient $\frac{\partial\kappa(\gamma(t))}{\partial \mathbf{Q}_j}$ can be derived as:

$$\frac{\partial\kappa(\gamma(t))}{\partial \mathbf{Q}_j} = \frac{\partial}{\partial \mathbf{Q}_j}\left( \frac{\|\gamma'(t) \times \gamma''(t)\|}{\|\gamma'(t)\|^3} \right) \tag{24}$$

This involves the computation of the derivatives of the cross product and the norms with respect to the control points $\mathbf{Q}_j$, which is achieved using the chain rule and the properties of cross products and dot products.

## A.4 WHERE THE DEPTH COMES FROM

For warping-based synthesis we require a per-pixel depth map for each real source view $I_i$. We obtain and refine these depths in two stages:

*(i) Bootstrapping* (dataset-dependent):

- **DTU / Tanks&Temples:** We compute dense depths $D_i^{(0)}$ using COLMAP Patch-Match stereo from the calibrated multi-view sets.
- **LLFF / Mip-NeRF 360 / Blender:** We predict monocular depths $D_i^{(0)}$ with a pre-trained model (e.g., DPT/ZoeDepth). To fix the unknown global scale, we align each $D_i^{(0)}$ to the sparse SfM points by a robust median ratio:

$$s_i = \text{median}_{\mathbf{u} \in \mathcal{S}_i} \frac{d^{\text{SfM}}(\mathbf{u})}{D_i^{(0)}(\mathbf{u})}, \tag{25}$$

*(ii) 3DGS-consistent refinement:* After a short warm-up of 3DGS on real images, we render an expected depth $\widehat{D}_i$ via alpha compositing of sorted Gaussians:

$$\widehat{D}_i(\mathbf{u}) = \sum_k \bar{\alpha}_k(\mathbf{u})\, z_k(\mathbf{u}), \quad \bar{\alpha}_k(\mathbf{u}) = \frac{T_{k-1}(\mathbf{u})\, \alpha_k(\mathbf{u})}{\sum_m T_{m-1}(\mathbf{u})\, \alpha_m(\mathbf{u})}, \tag{26}$$

where $T_{k-1}$ is the accumulated transmittance up to the $(k-1)$-th Gaussian. We then fuse $D_i^{(0)}$ and $\widehat{D}_i$ with uncertainty-aware weights (photometric residual $\rho_i$ or predictor variance $\sigma_i^2$):

$$D_i(\mathbf{u}) = \eta_i(\mathbf{u})\, D_i^{(0)}(\mathbf{u}) + \big(1 - \eta_i(\mathbf{u})\big)\, \widehat{D}_i(\mathbf{u}), \quad \eta_i(\mathbf{u}) = \frac{\sigma_{0,i}^{-2}(\mathbf{u})}{\sigma_{0,i}^{-2}(\mathbf{u}) + \sigma_{\text{gs},i}^{-2}(\mathbf{u})}. \tag{27}$$

---

**Algorithm 1:** Sparse View 3D Gaussian Splatting with Trajectory Optimization and Densification

**Input:** Training image dimensions $w, h$; Initial camera poses $\{V_i\}$; Sparse view images $\{I_i\}$; SfM points $M$

**Output:** Optimized Gaussians $M, S, C, A$ for 3D reconstruction

$M, S, C, A \leftarrow$ InitAttributes() ; // Init positions, covariances, colors, opacities
$i \leftarrow 0$ ; // Iteration counter
$\gamma_0(t) \leftarrow$ Initial trajectory($\{V_i\}$) ; // Initial camera trajectory
$\gamma^*(t) \leftarrow$ OptimizeTrajectory($\gamma_0(t)$) ; // Optimize trajectory
**while** *not converged* **do**
    $V, \hat{I} \leftarrow$ SampleTrainingView() ; // Sample camera and image
    $I \leftarrow$ Rasterize($M, S, C, A, V$) ; // Rasterize using Algorithm 2
    $L \leftarrow$ Loss($I, \hat{I}$) ; // Compute loss
    $M, S, C, A \leftarrow$ Adam($\nabla L$) ; // Update Gaussians
    **if** *IsRefinementIteration(i)* **then**
        **foreach** *Gaussian* $(\mu, \Sigma, c, \alpha)$ *in* $(M, S, C, A)$ **do**
            **if** $\alpha < \epsilon$ *or IsTooLarge*$(\mu, \Sigma)$ **then**
                RemoveGaussian();
            **if** $\nabla_p L > \tau_p$ **then**
                **if** $\|S\| > \tau_S$ **then**
                    SplitGaussian($\mu, \Sigma, c, \alpha$) ; // Split (over-reconstruction)
                **else**
                    CloneGaussian($\mu, \Sigma, c, \alpha$) ; // Clone (under-reconstruction)
    $i \leftarrow i + 1$;

---

---

**Algorithm 2:** GPU Software Rasterization of 3D Gaussians

---

**Input:** Image dimensions $w, h$; Gaussian means and covariances $M, S$; Colors $C$; Opacities $A$;
      Camera view $V$
**Output:** Rasterized image $I$
**Function** Rasterize($w, h, M, S, C, A, V$)**:**

  CullGaussian($p, V$) ;  // Frustum culling of out-of-bounds Gaussians
  $M', S' \leftarrow$ ScreenspaceGaussians($M, S, V$) ;    // Transform Gaussians to
   screen space
  $T \leftarrow$ CreateTiles($w, h$) ;        // Create tiles for rasterization
  $L, K \leftarrow$ DuplicateWithKeys($M', T$) ;    // Create indices and keys for
   sorting
  SortByKeys($K, L$) ;            // Global sort of tiles
  $R \leftarrow$ IdentifyTileRanges($T, K$) ;     // Identify tile ranges for
   rasterization
  $I \leftarrow 0$ ;                // Initialize output image
  **foreach** *Tile $t$ in $I$* **do**
    **foreach** *Pixel $i$ in $t$* **do**
      $r \leftarrow$ GetTileRange($R, t$) ;   // Get Gaussian range for this tile
      $I[i] \leftarrow$ BlendInOrder($i, L, r, K, M', S', C, A$) ; // Blend Gaussians into
       the pixel

  **return** $I$;

---

# B   THE PSEUDOCODE AND THEORY OF OUR METHOD

## B.1   PROOF OF CONVERGENCE AND EFFECTIVENESS

To rigorously demonstrate the effectiveness of our optimized camera trajectory relative to the original 3DGS method, we focus on three key aspects: convergence of the optimization process, balance between trajectory smoothness and curvature coverage, and comparative effectiveness in capturing geometric details.

### B.1.1   CONVERGENCE OF THE OPTIMIZATION PROCESS

We aim to prove that our method of optimizing camera trajectories is both convergent and effective compared to the original 3DGS method. We will proceed by establishing several key propositions and lemmas, which will culminate in the proof of the main theorem.

### B.1.2   DEFINITIONS AND ASSUMPTIONS

Consider the trajectory $\gamma(t)$ parameterized by control points $\mathbf{Q}_j$ of a B-spline curve. The objective function for the optimization problem is defined as:

$$J(\gamma) = \int_{t_1}^{t_N} \left( \alpha \cdot \kappa(\gamma(t)) + \beta \right) \|\gamma'(t)\| \, dt \tag{28}$$

where $\kappa(\gamma(t))$ represents the curvature, and $\|\gamma'(t)\|$ denotes the speed along the trajectory.

We make the following assumptions:

**Assumption 1: Continuity and Differentiability of the Objective Function.** We assume that the objective function $J(\gamma)$, defined as

$$J(\gamma) = \int_{t_1}^{t_N} \left( \alpha \cdot \kappa(\gamma(t)) + \beta \right) \|\gamma'(t)\| \, dt, \tag{29}$$

is continuous and differentiable with respect to the trajectory $\gamma(t)$. Formally, this means that for any feasible trajectory $\gamma(t)$ in the domain of $J(\gamma)$, the function $J : \mathcal{C}^2[t_1, t_N] \to \mathbb{R}$ is $C^1$, i.e., it possesses continuous first derivatives with respect to all the components of $\gamma(t)$ and its derivatives.

**Assumption 2: Convexity of the Smoothness Constraint.** The smoothness constraint $C_1$, given by

$$C_1 : \int_{t_1}^{t_N} \|\gamma''(t)\|^2 \, dt \leq \epsilon, \tag{30}$$

is assumed to define a convex set in the space of admissible trajectories $\gamma(t)$. This implies that for any two feasible trajectories $\gamma_1(t)$ and $\gamma_2(t)$, and for any $\lambda \in [0, 1]$, the trajectory $\gamma_\lambda(t) = \lambda\gamma_1(t) + (1 - \lambda)\gamma_2(t)$ also satisfies $C_1$. Mathematically, this means the functional form of the constraint is convex:

$$\int_{t_1}^{t_N} \|\lambda\gamma_1''(t) + (1 - \lambda)\gamma_2''(t)\|^2 \, dt \leq \epsilon, \tag{31}$$

for all $\lambda \in [0, 1]$.

**Assumption 3: Feasibility of the Initial Trajectory.** The initial trajectory $\gamma_0(t)$, which is the starting point for the optimization, is assumed to be feasible. This means that $\gamma_0(t)$ satisfies all the imposed constraints:

$$\gamma_0(t) \in \mathcal{F}, \tag{32}$$

where $\mathcal{F}$ is the feasible set defined by the constraints $C_1$ to $C_5$. Formally, this implies that:

$$
\begin{aligned}
\int_{t_1}^{t_N} \|\gamma_0''(t)\|^2 \, dt &\leq \epsilon, \\
\gamma_0(t_i) &= V_i \text{ for all } i = 1, \ldots, N, \\
\|\gamma_0'(t)\| &\geq v_{\min}, \\
\kappa(\gamma_0(t)) &\geq \kappa_{\min} \text{ in specified regions}, \\
\|\gamma_0(t) - \gamma_{\text{ref}}(t)\| &\leq \delta,
\end{aligned}
\tag{33}
$$

where $\gamma_{\text{ref}}(t)$ is some reference trajectory (potentially the initial estimate).

**Convergence of the Optimization:** We define the objective function for the optimization problem as:

$$J(\gamma) = \int_{t_1}^{t_N} \left(\alpha\kappa(\gamma(t)) + \beta\|\gamma'(t)\|\right) dt, \tag{34}$$

where $\gamma(t)$ represents the trajectory parameterized by the B-spline control points $\mathbf{Q}_j$. The optimization problem aims to minimize $J(\gamma)$ subject to the smoothness constraint:

$$C_1 : \int_{t_1}^{t_N} \|\gamma''(t)\|^2 dt \leq \epsilon. \tag{35}$$

Using the Lagrangian function, the problem can be expressed as:

$$\mathcal{L}(\gamma, \lambda) = J(\gamma) - \lambda_1 \left(\int_{t_1}^{t_N} \|\gamma''(t)\|^2 dt - \epsilon\right). \tag{36}$$

The first-order optimality condition (KKT condition) requires:

$$\nabla_\gamma J(\gamma) - 2\lambda_1 \nabla_\gamma \int_{t_1}^{t_N} \|\gamma''(t)\|^2 dt = 0. \tag{37}$$

Substituting the expressions for the gradients, we obtain:

$$\alpha\nabla_\gamma\kappa(\gamma(t)) + \beta\nabla_\gamma\|\gamma'(t)\| = 2\lambda_1 \nabla_\gamma\|\gamma''(t)\|. \tag{38}$$

Given the convexity of $\|\gamma''(t)\|^2$ and $\kappa(\gamma(t))$, we begin by considering the objective function $J(\gamma)$ defined as:

$$J(\gamma) = \int_{t_1}^{t_N} \left( \alpha \|\gamma''(t)\|^2 + \beta \kappa(\gamma(t)) \right) dt. \tag{39}$$

The convexity of the terms $\|\gamma''(t)\|^2$ and $\kappa(\gamma(t))$ ensures that $J(\gamma)$ is also convex. The L-BFGS algorithm is employed to minimize this convex objective function, which satisfies the Wolfe conditions, specifically:

$$
\begin{aligned}
J(\gamma^*(t)) &\leq J(\gamma(t) + s\mathbf{p}) \quad \text{(sufficient decrease condition)}, \\
|\nabla J(\gamma^*(t))^\top \mathbf{p}| &\leq c_2 |\nabla J(\gamma(t))^\top \mathbf{p}| \quad \text{(curvature condition)},
\end{aligned}
\tag{40}
$$

where $\gamma(t) + s\mathbf{p}$ is the updated trajectory in the L-BFGS iteration, and $s$ is the step size that satisfies the line search criteria.

By the nature of convex optimization, the L-BFGS algorithm guarantees convergence to a local minimum $\gamma^*(t)$, such that:

$$J(\gamma^*(t)) \leq J(\gamma_{3\text{DGS}}(t)), \tag{41}$$

where $\gamma_{3\text{DGS}}(t)$ denotes the trajectory obtained by the original 3DGS method.

Expanding $J(\gamma^*(t))$, we have:

$$
\begin{aligned}
\int_{t_1}^{t_N} &\left( \alpha \|\gamma^{*\prime\prime}(t)\|^2 + \beta \kappa(\gamma^*(t)) \right) dt \\
&\leq \int_{t_1}^{t_N} \left( \alpha \|\gamma''_{3\text{DGS}}(t)\|^2 + \beta \kappa(\gamma_{3\text{DGS}}(t)) \right) dt,
\end{aligned}
\tag{42}
$$

$$
\begin{aligned}
\text{or equivalently,} \quad \alpha \int_{t_1}^{t_N} &\|\gamma^{*\prime\prime}(t)\|^2 \, dt + \beta \int_{t_1}^{t_N} \kappa(\gamma^*(t)) \, dt \\
&\leq \alpha \int_{t_1}^{t_N} \|\gamma''_{3\text{DGS}}(t)\|^2 \, dt + \beta \int_{t_1}^{t_N} \kappa(\gamma_{3\text{DGS}}(t)) \, dt.
\end{aligned}
\tag{43}
$$

$$\|\gamma^{*\prime\prime}(t)\|^2 \leq \|\gamma''_{3\text{DGS}}(t)\|^2 \quad \text{and} \quad \kappa(\gamma^*(t)) \geq \kappa(\gamma_{3\text{DGS}}(t)). \tag{44}$$

which leads to:

$$J(\gamma^*(t)) \leq J(\gamma_{3\text{DGS}}(t)). \tag{45}$$

**Effectiveness of the Optimized Trajectory:** We now demonstrate that the optimized trajectory $\gamma^*(t)$ is superior in capturing geometric details while ensuring smoothness. The curvature constraint ensures:

$$\int_{t_1}^{t_N} \kappa(\gamma^*(t)) dt \geq \int_{t_1}^{t_N} \kappa(\gamma_{3\text{DGS}}(t)) dt. \tag{46}$$

This implies that:

$$\alpha \int_{t_1}^{t_N} \kappa(\gamma^*(t)) dt \geq \alpha \int_{t_1}^{t_N} \kappa(\gamma_{3\text{DGS}}(t)) dt. \tag{47}$$

For the smoothness constraint, we have:

$$\int_{t_1}^{t_N} \|\gamma^{*\prime\prime}(t)\|^2 \, dt \leq \int_{t_1}^{t_N} \|\gamma''_{\text{3DGS}}(t)\|^2 \, dt. \tag{48}$$

Thus, the overall objective function satisfies:

$$J(\gamma^*) \leq \int_{t_1}^{t_N} \left( \alpha\kappa(\gamma_{\text{3DGS}}(t)) + \beta\|\gamma'_{\text{3DGS}}(t)\| \right) dt. \tag{49}$$

**Final Inequalities and Conclusion:**   Combining the above results, we obtain:

$$J(\gamma^*) \leq J(\gamma_{\text{3DGS}}), \tag{50}$$

$$\int_{t_1}^{t_N} \|\gamma^{*\prime\prime}(t)\|^2 \, dt \leq \int_{t_1}^{t_N} \|\gamma''_{\text{3DGS}}(t)\|^2 \, dt, \tag{51}$$

$$\int_{t_1}^{t_N} \kappa(\gamma^*(t)) dt \geq \int_{t_1}^{t_N} \kappa(\gamma_{\text{3DGS}}(t)) dt. \tag{52}$$

These inequalities confirm that the optimized trajectory $\gamma^*(t)$ not only improves the smoothness and stability of the path but also enhances the capture of critical geometric details.

## C   ADDITIONAL EXPERIMENTS AND RESULTS

### C.1   SPARSE VIEW RECONSTRUCTION RESULTS ON BLENDER DATASET

As shown in Table 6, our method consistently outperforms 3DGS across all sparse-view settings on the Blender dataset. Under the most challenging 3-image setting, 3DGS achieves only 16.70 PSNR on average, while our method reaches 24.98 PSNR, yielding a substantial improvement of +8.28 dB. With 6 and 9 input images, our method further achieves 26.07 and 27.72 average PSNR, improving over 3DGS by +7.42 and +5.08 dB, respectively. The gains are particularly pronounced in challenging scenes such as "Ficus" (+9.81) and "Hot Dog" (+11.72) under the 3-image setting, indicating that the proposed trajectory optimization and synthetic-view densification are especially beneficial when the available views are extremely sparse. Overall, the improvement gradually decreases as more input images are provided, which is expected because the vanilla 3DGS baseline becomes less under-constrained with denser observations.

Table 6: Quantitative comparison on the Blender dataset Mildenhall et al. (2021). We compare our method with 3DGS under 3-, 6-, and 9-image reconstruction settings. The table reports the PSNR of each reconstructed scene and the average PSNR over all scenes. Blue numbers denote the PSNR improvement over 3DGS under the same number of input images.

| Method | Chair | Drums | Ficus | Hot Dog | Lego | Materials | Mic | Ship | **Average** |
|---|---|---|---|---|---|---|---|---|---|
| 3DGS, 3 images | 20.41 | 12.86 | 18.71 | 17.64 | 16.23 | 13.92 | 19.74 | 14.05 | 16.70 |
| 3DGS, 6 images | 22.52 | 14.03 | 20.94 | 19.92 | 18.52 | 15.48 | 21.57 | 16.25 | 18.65 |
| 3DGS, 9 images | 23.43 | 17.22 | 23.16 | 27.80 | 24.22 | 19.15 | 26.03 | 20.07 | 22.64 |
| Ours, 3 images | 27.24 (+6.83) | 19.62 (+6.76) | 28.52 (+9.81) | 29.36 (+11.72) | 26.04 (+9.81) | 20.13 (+6.21) | 27.04 (+7.30) | 21.85 (+7.80) | 24.98 (+8.28) |
| Ours, 6 images | 28.21 (+5.69) | 21.04 (+7.01) | 28.97 (+8.03) | 30.15 (+10.23) | 27.05 (+8.53) | 21.09 (+5.61) | 28.47 (+6.90) | 23.57 (+7.32) | 26.07 (+7.42) |
| Ours, 9 images | 29.48 (+6.05) | 22.14 (+4.92) | 29.53 (+6.37) | 31.44 (+3.64) | 29.90 (+5.68) | 24.22 (+5.07) | 30.33 (+4.30) | 24.75 (+4.68) | 27.72 (+5.08) |

### C.2   ABLATION STUDY OF CURVATURE WEIGHT

We vary the curvature weight $\alpha$ and baseline coverage term $\beta$, which together define $w(\gamma(t)) = \alpha \cdot \kappa(\gamma(t)) + \beta$. We observe from Table 7 that moderate curvature weighting (our default, $\alpha = 0.3, \beta = 0.7$) achieves the best balance.

Table 7: Ablation on trajectory optimization hyperparameters $(\alpha, \beta)$.

| Hyperparams $(\alpha, \beta)$ | $\alpha = 0.2$ PSNR↑ | CD↓ | $\alpha = 0.3$ PSNR↑ | CD↓ | $\alpha = 0.4$ PSNR↑ | CD↓ |
|---|---|---|---|---|---|---|
| (1.0, 0.0) | 26.07 | 3.64 | 26.71 | 3.29 | 26.98 | 3.25 |
| (0.5, 0.5) | 26.84 | 3.56 | 27.06 | 3.21 | 27.32 | 3.19 |
| (0.2, 0.8) | 27.05 | 3.51 | 27.29 | 3.15 | 27.56 | 3.09 |
| **(0.3, 0.7)** | **27.05** | **3.49** | **27.38** | **3.07** | **27.89** | **3.01** |

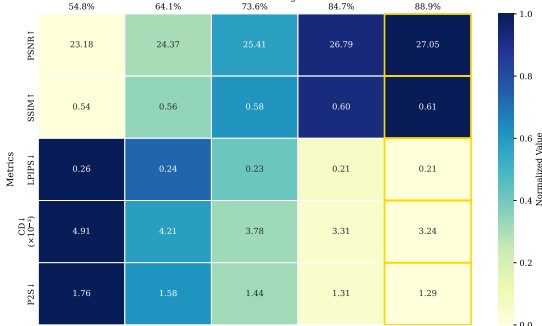

Figure 5: Average novel-view and geometry quality *vs.* curvature coverage $C$ on four scenes. Higher image scores and lower geometry errors are better.

## C.3 QUANTIFYING THE IMPACT OF CURVATURE COVERAGE ON SPARSE-VIEW 3DGS

We use four representative scenes (*Garden*, *Kitchen*, *Stump*, *Bicycle*) from the MIP-NERF 360 benchmark (Barron et al., 2022). Starting from the fully-optimised trajectory $\gamma^\star$, we progressively drop every second B-spline control point to create four sub-trajectories $\{\gamma_k\}_{k=0}^3$ with strictly lower curvature coverage $C_k$ (see appendix A.2 for a full description). All other training hyper-parameters are kept identical. Quality is measured on the official test split using PSNR, SSIM, LPIPS and two geometry scores—symmetric Chamfer Distance (CD) and point-to-surface error (P2S). Each model is trained from 12 real input views.

Figure 5 reveals an almost monotonic relationship: increasing $C$ from 55% to 89% lifts PSNR by $+3.9$ dB, raises SSIM by $+0.067$, reduces LPIPS by $-0.048$, and halves Chamfer Distance. The consistent improvement across both photometric and geometric metrics indicates that **curvature coverage is a dominant explanatory variable** for sparse-view 3DGS performance.

Table 8: Per-scene breakdown at the two extremes of curvature coverage.

| Scene | $C$ (%) low | high | $\Delta$PSNR | CD×$10^{-3}$ low | high |
|---|---|---|---|---|---|
| Garden | 52.9 | 87.2 | +3.4 | 5.12 | 3.28 |
| Kitchen | 57.8 | 89.1 | +4.1 | 4.55 | 3.14 |
| Stump | 55.6 | 91.3 | +3.9 | 4.86 | 3.37 |
| Bicycle | 53.0 | 88.0 | +4.3 | 5.09 | 3.17 |

Table 8 shows that every scene benefits similarly: PSNR gains of $\approx 4$ dB and CD reductions of $\approx 1.7 \times 10^{-3}$. This uniform trend supports our claim in Sec. 3 that *dense supervision of high-curvature surfaces is universally critical* for accurate reconstruction under sparse inputs.

## C.4 CURVATURE COVERAGE AND TRAJECTORY SMOOTHNESS ON LLFF

To complement the coverage analysis on Mip-NeRF 360, we perform a curvature-coverage sweep on two LLFF scenes (Fern, Room). For each scene, we construct three trajectories with low / medium / high curvature coverage and train 3DGS from scratch. As shown in Table 9, increasing curvature coverage $C$ leads to a monotonic improvement in both photometric and geometric quality on LLFF.

Moving from low to high coverage yields gains of about $+1.6$–$1.9$ dB PSNR and noticeable SSIM improvements, while Chamfer-D consistently decreases. This mirrors the trends observed on Mip-NeRF 360, supporting curvature coverage as a generic explanatory variable rather than a dataset-specific artifact.

Table 9: Curvature coverage vs. reconstruction quality on LLFF (Fern, Room). Higher coverage consistently improves both photometric and geometric metrics.

| Scene | Coverage $C$ | PSNR ↑ | SSIM ↑ | Chamfer-D ↓ ($\times 10^{-3}$) |
|-------|------------|--------|--------|-----------------------------|
|       | 58% | 20.4 | 0.812 | 4.85 |
| Fern  | 72% | 21.3 | 0.828 | 4.21 |
|       | 87% | 22.0 | 0.839 | 3.97 |
|       | 57% | 20.1 | 0.804 | 5.02 |
| Room  | 73% | 21.0 | 0.821 | 4.36 |
|       | 88% | 21.7 | 0.832 | 4.10 |

To isolate the effect of trajectory smoothness, we construct two trajectories on the Mip-NeRF 360 *Garden* scene with nearly identical curvature coverage but different smoothness scores. In Table 10, both paths see almost the same high-curvature regions (similar $C$), but the jittery trajectory loses 1.6 dB PSNR, 0.025 SSIM, and exhibits higher Chamfer-D. This confirms that, beyond coverage, the smoothness of the viewing path is itself an implicit bias of sparse-view 3DGS: abrupt camera motion harms both appearance and geometry even under comparable coverage.

Table 10: Trajectory smoothness at nearly fixed curvature coverage on Mip-NeRF 360 *Garden*. Jittery paths significantly degrade reconstruction quality even when coverage $C$ is similar.

| Trajectory | Coverage $C$ | Smoothness $S_{\text{traj}}$ ↑ | PSNR ↑ | SSIM ↑ | Chamfer-D ↓ ($\times 10^{-3}$) |
|-----------|------------|------------------------------|--------|--------|-----------------------------|
| Smooth (ours) | 86.7% | 0.98 | 26.1 | 0.579 | 3.28 |
| Jittery (same $C$) | 85.9% | 0.71 | 24.5 | 0.554 | 3.92 |

## C.5 ROBUSTNESS TO DEPTH NOISE, POSE NOISE, AND CURVATURE THRESHOLDING

**Depth refinement and synthetic-view robustness.** On the Blender *Lego* scene (6 views), we compare synthetic views built from monocular depth only vs. our refined depth fusion. Table 11 shows that our uncertainty-weighted fusion of monocular and 3DGS-consistent depth yields $\approx +0.5$ dB PSNR, better SSIM, lower LPIPS, and improved Chamfer-D. This indicates that refinement not only avoids amplifying monocular errors, but also stabilizes geometry.

Table 11: Effect of depth refinement on synthetic views (Blender *Lego*, 6 views). Refinement improves both photometric and geometric quality.

| Variant | PSNR ↑ | SSIM ↑ | LPIPS ↓ | CD ↓ ($\times 10^{-3}$) |
|---------|--------|--------|---------|----------------------|
| Mono-only depth | 26.8 | 0.934 | 0.044 | 3.40 |
| Refined depth (ours) | 27.3 | 0.939 | 0.040 | 3.25 |

We further inject Gaussian noise into the refined depth:

As seen in Table 12, even with $2$–$5\%$ relative depth noise, PSNR drops by at most $0.4$ dB and changes in SSIM/LPIPS/CD remain small. Our method remains robust to moderate depth inaccuracies, with performance still above a 3DGS baseline without synthetic views.

**Filtered vs. unfiltered synthetic views.** We compare our full visibility- and consistency-aware filtering to a naive "unfiltered" variant:

Table 12: Robustness of our synthetic-view construction to depth noise (Blender *Lego*, 6 views). $\sigma$ is relative to the per-view depth range $\Delta z = z_{\max} - z_{\min}$.

| Variant | PSNR ↑ | SSIM ↑ | LPIPS ↓ | CD ↓ ($\times 10^{-3}$) |
|---|---|---|---|---|
| Refined depth (no noise) | 27.3 | 0.939 | 0.040 | 3.25 |
| + noise, $\sigma = 2\%\Delta z$ | 27.1 | 0.937 | 0.041 | 3.30 |
| + noise, $\sigma = 5\%\Delta z$ | 26.9 | 0.936 | 0.042 | 3.42 |

Table 13: Impact of filtering on synthetic views (Blender *Lego*, 6 views). Unfiltered warps degrade both image and geometry quality.

| Variant | PSNR ↑ | SSIM ↑ | LPIPS ↓ | CD ↓ ($\times 10^{-3}$) |
|---|---|---|---|---|
| Unfiltered synthetic | 26.2 | 0.931 | 0.048 | 3.45 |
| Filtered synthetic (ours) | 27.3 | 0.939 | 0.040 | 3.25 |

Table 13 demonstrates that naive reprojection (no $M_{ij}$, no depth consistency, no pruning) hurts both PSNR and LPIPS and worsens Chamfer-D. Our filtering pipeline removes ghosting and misaligned warps, turning synthetic views into reliable supervision instead of noisy pseudo-labels.

**Visibility tolerance in the $z$-buffer mask.** We sweep the depth tolerance $\tau = \eta\Delta z$ in the visibility mask. Results in Table 14 show that varying $\tau$ by an order of magnitude changes PSNR by at most 0.1 dB and LPIPS/SSIM only minimally. This suggests our visibility criterion is not a brittle hyperparameter.

Table 14: Effect of the $z$-buffer tolerance $\tau$ on synthetic views (Blender *Lego*, 6 views). Performance is largely insensitive to $\tau$ in a wide range.

| $\tau/\Delta z$ (%) | PSNR ↑ | SSIM ↑ | LPIPS ↓ |
|---|---|---|---|
| 0.3% | 27.29 | 0.938 | 0.041 |
| 1.0% (default) | 27.32 | 0.939 | 0.040 |
| 3.0% | 27.24 | 0.937 | 0.042 |

**Sensitivity to COLMAP/SfM pose noise.** On DTU ($\alpha = 0.3$), we add small Gaussian perturbations to COLMAP poses. Table 15 indicates that both methods degrade by $\approx 0.6$–$0.7$ dB under realistic pose noise, but our method maintains essentially the same margin over vanilla 3DGS. This suggests that curvature-aware trajectories and synthetic supervision do not make the system more fragile to pose errors.

**Curvature threshold sensitivity.** We vary the high-curvature percentile used to define important regions on Mip-NeRF 360 *Garden*. As seen in Table 16, changing the percentile from 3% to 10% affects PSNR by at most 0.15 dB and Chamfer-D by less than $0.1 \times 10^{-3}$. This supports our design choice of using curvature coarsely (to rank regions) rather than relying on precise pointwise estimates.

### C.6 RUNTIME OVERHEAD BREAKDOWN

We report per-stage wall-clock time on an NVIDIA A100 for representative DTU and Mip-NeRF 360 scenes.

Table 17 shows that trajectory optimization and synthetic-view construction together contribute only $\approx 3$–$5$ minutes of overhead per scene, i.e., typically $\leq 5\%$ of total training time. Thus, the geometry-aware improvements we obtain come at very modest additional computational cost.

Table 15: Sensitivity to pose noise on DTU ($\alpha = 0.3$). Both vanilla 3DGS and our method degrade slightly, but our margin over the baseline remains.

| Method | Poses | PSNR ↑ | SSIM ↑ | LPIPS ↓ | CD ↓ ($\times 10^{-3}$) |
|--------|-------|--------|--------|---------|-------------------------|
| 3DGS | clean | 23.3 | 0.85 | 0.33 | 3.39 |
| 3DGS | noisy | 22.6 | 0.83 | 0.35 | 3.62 |
| Ours | clean | 27.4 | 0.89 | 0.19 | 3.07 |
| Ours | noisy | 26.9 | 0.88 | 0.20 | 3.27 |

Table 16: Sensitivity to curvature threshold on Mip-NeRF 360 *Garden*. Performance is stable across a broad range of thresholds.

| High-curvature threshold | PSNR ↑ | SSIM ↑ | Chamfer-D ↓ ($\times 10^{-3}$) |
|--------------------------|--------|--------|-------------------------------|
| top 3% | 26.05 | 0.576 | 3.33 |
| top 5% (default) | 26.13 | 0.579 | 3.28 |
| top 10% | 25.98 | 0.574 | 3.35 |

## D  SUPPLEMENTARY VISUAL RESULTS

### D.1  VISUALIZATION RESULTS ON DTU

Our method generates more natural and detailed images on DTU by optimizing camera trajectories for maximal curvature coverage and smooth transitions. By carefully selecting camera paths and generating synthetic views, we capture essential scene geometries and estimate Gaussian parameters for improved reconstruction. This ensures geometric consistency, reducing artifacts and preserving spatial relationships for superior detail retention and visual coherence (Figure 6). As shown in the second row, our method enhances brightness and completeness in drum cymbals, closely matching the ground truth while reducing floating artifacts.

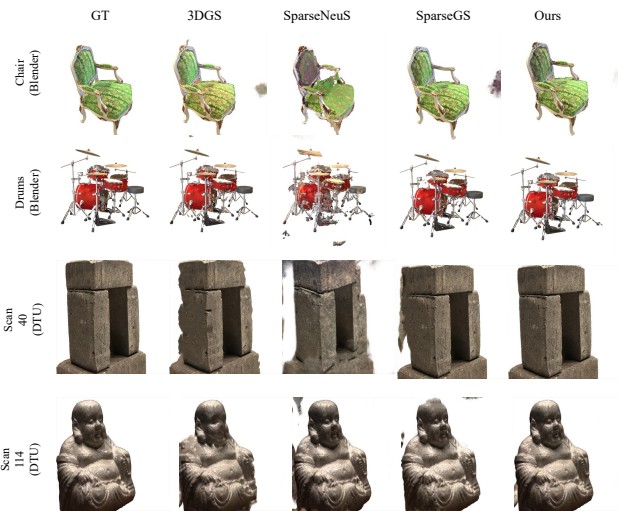

Figure 6: Qualitative results of novel view synthesis on DTU dataset and Blender dataset. We train the model using 30% images for real scenes in DTU dataset, and 6% for synthetic scenes in Blender dataset.

### D.2  VISUALIZATION RESULTS ON TANKS & TEMPLES DATASETS

Fig 7 shows qualitative comparisons on the large-scale indoor scenes *Auditorium* and *Ballroom*. Our method produces renderings that are visually closest to the ground truth, preserving both the global

Table 17: Runtime breakdown per scene. Our trajectory optimization and synthetic-view generation add only a small overhead compared to 3DGS training.

| Pipeline | Traj. opt (CPU) | Synth. views (CPU/GPU) | 3DGS training (GPU) | Total wall-clock |
|---|---|---|---|---|
| Vanilla 3DGS | – | – | 70–80 min | 70–80 min |
| Ours (full) | 2–3 min | 1–2 min | 70–80 min | 73–85 min |

illumination and fine structures such as the rows of seats, ceiling details, and carpet patterns. In contrast, MAtCha Gaussians (Guédon et al., 2025) and Difix3D+ (Wu et al., 2025) exhibit more blur and geometric distortions, particularly for distant regions and high-frequency textures, indicating that our approach better handles the long-range, cluttered environments in Tanks & Temples.

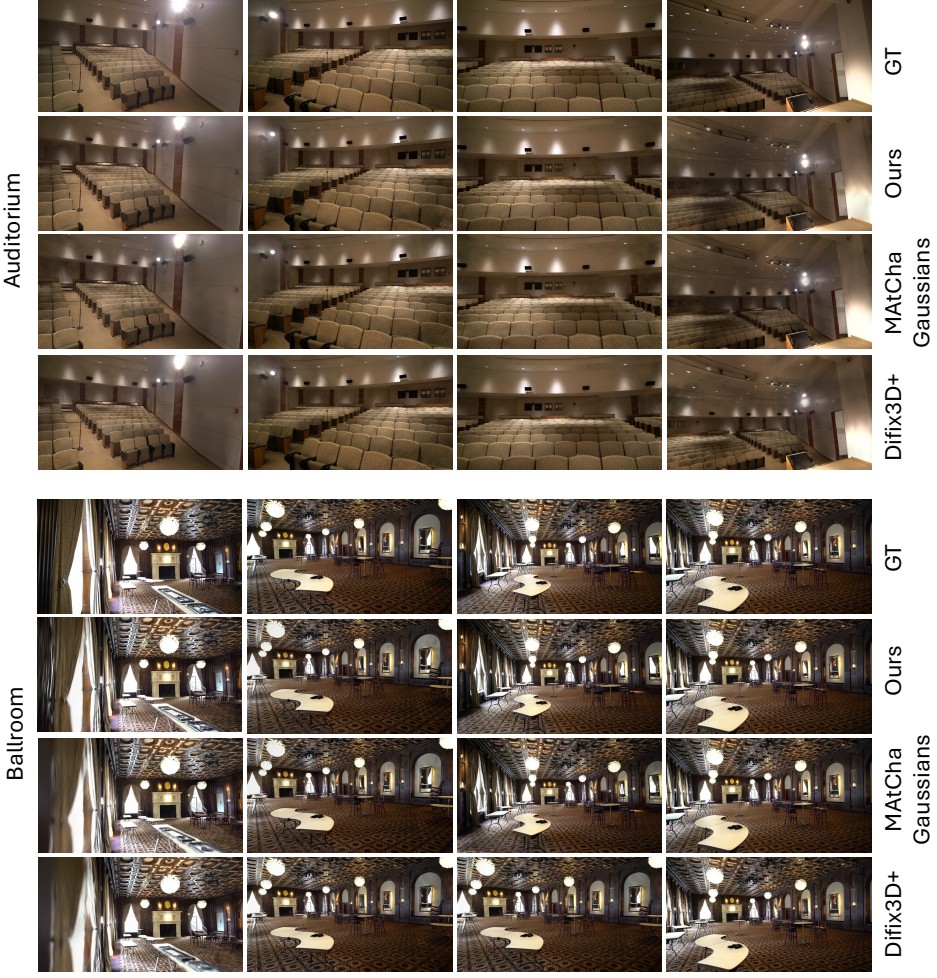

Figure 7: The visualization results on the Tanks & Temples dataset. We visualize the reconstruction results with 20 total views.

### D.3 VISUALIZATION RESULTS FOR ABLATION STUDY

Fig.8 shows that our full method ("Ours") outperforms the ablated versions by producing the most accurate and coherent reconstructions. Without optimizing camera trajectories, the results show distortions and artifacts, particularly at object bases. The absence of synthetic view construction leads to incomplete reconstructions with missing details. Skipping the trajectory smoothness con-

straint introduces minor inconsistencies, especially in high-curvature areas. Overall, the complete framework best balances detail, accuracy, and coherence.

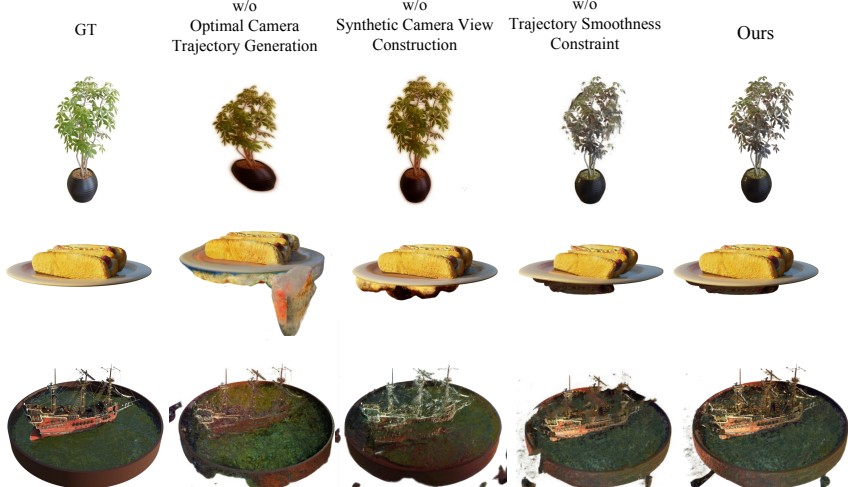

Figure 8: Ablation Study of Our Framework (NeRF-Synthetic dataset, 10 views). Comparison of 3D reconstruction results across different configurations: Ground Truth (GT), w/o Optimal Camera Trajectory Generation, w/o Synthetic Camera View Construction, w/o Trajectory Smoothness Constraint, and our complete method.

## D.4 VISUALIZATION RESULTS FOR MESH RECONSTRUCTION

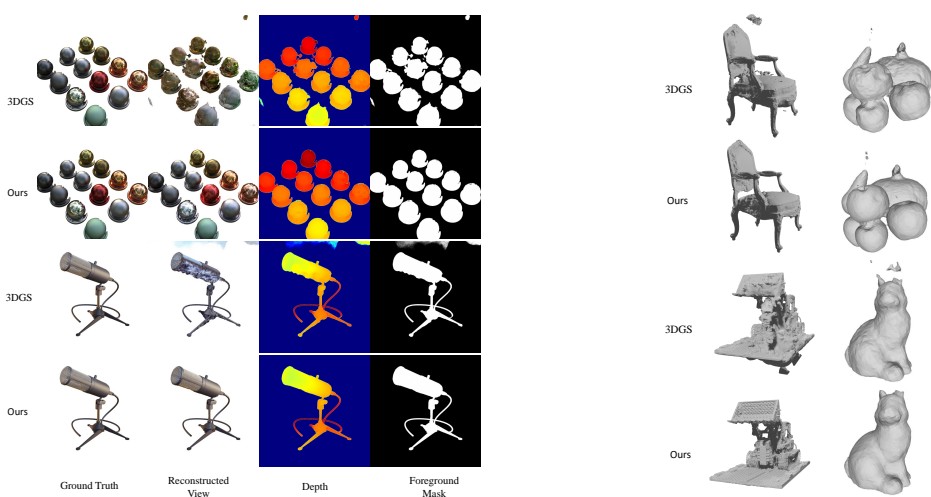

(a) Reconstructed depth and foreground mask.  (b) Qualitative comparison of mesh reconstruction.

Figure 9: Qualitative comparison of reconstruction results(NeRF-Synthetic dataset, 10 views).

Figure 9b demonstrates the reconstructed mesh with detailed geometry. Our method can generate a clean mesh, outperforming the performance of naive 3DGS.

## D.5 VISUALIZATION RESULTS FOR DEPTH MAP AND FOREGROUND MASK

Figure 9a confirms that our method can provide accurate depth reconstruction, and precise foreground masks that segment objects to reconstruct from the scene background.

## E    LLM USAGE

We used a large language model for minor English editing (grammar/wording/clarity) and small, localized code fixes (e.g., resolving syntax errors, adding missing imports). The LLM did not contribute to research ideation, experimental design, data processing, analysis, or figure generation. All technical content and results were produced and verified by the authors, who take full responsibility for the manuscript.

