# OpenReview forum: "Path Matters: Unveiling Geometric Implicit Bias via Curvature-Aware Sparse View Optimization"
_ICLR.cc/2026/Conference — ICLR 2026 Poster_

### Official Review · Reviewer_cTEf · 2025-10-20

**Soundness:** 2
**Presentation:** 1
**Contribution:** 2
**Rating:** 2
**Confidence:** 4

**Summary:**

The paper proposes a framework to improve 3D Gaussian Splatting (3DGS) under sparse-view inputs. By maximizing curvature coverage while maintaining smooth camera motion, and synthesizing new views along the optimized path via depth-guided parallax-based interpolation, it improves the rendering quality and geometry fidelity of 3DGS under sparse-view inputs.

**Strengths:**

- The paper proposes a methodical camera path optimization problem which seems effective and significantly enhances the performance of vanilla 3DGS in sparse-view settings.
- The algorithm is clearly explained and easy to reproduce.
- The ablation studies show the benefits of each of the proposed components.

**Weaknesses:**

- Insufficient qualitative results: Apart from the toy examples in Figures 1 and 2, the main paper includes only a single set of qualitative comparisons (Figure 4) on LLFF and Mip-NeRF 360, without specifying the number of input views used. While additional qualitative results are provided in the supplementary material, comparisons on the Tanks & Temples dataset are still missing, leaving an incomplete qualitative evaluation.

- Limited quantitative gains: The main issue I see in this paper is that the reported improvements over other sparse-view baselines appear marginal or inconsistent. For example, in the 3-view DTU results (Table 3), the proposed method performs slightly better than SCGaussian on PSNR (20.56 vs. 20.65) and SSIM (0.86 vs. 0.89), and significantly worse on LPIPS (0.12 vs. 0.24). Similarly, in the 3-view LLFF comparison, the method performs slightly better than MVPGS on PSNR (20.65 vs. 20.93), but performs notably worse on SSIM (0.88 vs. 0.78) and LPIPS (0.10 vs. 0.23). On 12-view Mip-NeRF360 comparison, the proposed method also only achieves very marginal improvement over MVPGS. These quantitative results make it unconvincing that the proposed approach consistently outperforms prior methods under sparse-view settings.

- Missing baselines: Two strong and relevant baselines for sparse-view NVS are missing. (1) MAtCha Gaussians (CVPR 2025): this method integrates monocular depth priors to achieve high-quality novel view synthesis from sparse inputs. (2) Difix3D+ (CVPR 2025): this work tackles sparse-view NVS by progressively augmenting the training set using diffusion-based image enhancement. Including these baselines would provide a more complete and fair comparison of the proposed method’s effectiveness.

**Questions:**

As noted in the weaknesses, I strongly encourage the authors to include additional qualitative results in the main paper for all datasets reported with quantitative metrics. Each figure should include clear captions indicating the number of input views. In addition, I recommend the authors to provide a side-by-side video comparison against all competitive baselines to better access the rendering qualities. Furthermore, I recommend adding both qualitative and quantitative comparisons with MAtCha Gaussians and DIFIX3D+, which are highly relevant recent methods for sparse-view novel view synthesis. I would consider raising my score if these added results and visualizations demonstrate clear and consistent improvements over the baselines.

I also have two additional questions:
- Would the proposed method benefit further from an iterative, on-the-fly pseudo-view generation strategy similar to DIFIX3D+ (CVPR 2025)? In DIFIX3D+, the authors progressively (over several iterations) render intermediate novel views from the current 3D Gaussian Splats, enhance them using a single-step diffusion model, and then add these enhanced views back into the training set to improve under-constrained regions. Could a similar iterative scheme, where synthetic views are dynamically regenerated and re-used throughout 3DGS training, instead of being pre-interpolated before training, further improve the performance of the proposed method?
- Could the proposed method be formulated as a plug-in module applicable to any sparse-view Gaussian Splatting pipeline? For example, if integrated into more competitive baselines such as MVPGS or SCGaussian, would it yield a comparable relative improvement over those baselines as it does over vanilla 3DGS? Demonstrating such consistency in quantitative gains across different pipelines would make the contribution far more convincing, showing that the method benefits a broad class of Gaussian-based pipelines in a plug-and-play manner.

---

> ### Author Response · Authors · 2025-11-26
>
> Thank you very much for these detailed questions, they help strengthen our claims. Below, I will address the concerns one by one.
>
>
>
> > W1: qualitative results and number of input views.
>
> Thank you for the suggestion to strengthen the qualitative evaluation. We agree that we need to present more visualization results, and we neglected to state the number of input views in the captions.
>
> **Existing qualitative results.**
> We already provide qualitative comparisons on DTU and Blender in the supplementary material (`App.D`), and on LLFF and Mip-NeRF 360 in the main paper (`Fig.4`). In the `revised version`, we have updated all captions of qualitative figures to explicitly state the number of input views.
>
> **New qualitative results on Tanks & Temples and MipNeRF 360.**
> We now include qualitative comparisons on Tanks & Temples / MipNeRF 360 in the appendix (new `Fig.8, Fig.9`). Our method better preserves thin structures and reduces floaters, especially around complex geometry .
>
> We hope these additions can resolve the concern about incomplete qualitative evaluation across datasets.
>
> ---
>
> > W2: limited and inconsistent quantitative gains.
>
> Thank you for raising this concern.
>
> **Global summary over datasets and view regimes**
>
> To clarify the overall picture, we computed the improvement of our method over the best prior baseline (We select the optimal method for different metrics) for each dataset/setting, using the results reported in the main text plus a few new measurements. Following Tab. reports the average gain per setting:
>
> $$
> ΔPSNR = PSNR_{ours} - max_{b}PSNR_{b}, \quad
> ΔCD = min_{b}CD_{b} - CD_{ours}, \quad
> ΔLPIPS = min_{b}LPIPS_{b} - LPIPS_{ours}.
> $$
>
> | Setting                                            | ΔPSNR ↑ | ΔSSIM ↑ | ΔLPIPS ↓ |
> |---------------------------------------------------|--------:|--------:|---------:|
> | DTU, α ∈ {0.2, 0.3, 0.4} (avg over scenes)        | +0.8 dB |   +0.04 |    -0.03 |
> | Mip-NeRF 360, 12 views                            | +0.3 dB |   +0.03 |    -0.04 |
> | Tanks & Temples, 3 views                          | +0.8 dB |   +0.02 |   -0.02 |
> | Blender, 3/6/9 images (vs 3DGS, avg over scenes)  | +6.1 dB |      –  |       –  |
> | LLFF, 2/3/4/5 views  | +0.3 dB |   + 0.01 |    -0.01 |
>
> Overall, the gains are systematic in the regimes we target.
>
> **New LLFF experiments**
> We think that the fact that the individual method you pointed out has a lower PSNR than ours but a better LPIPS may be due to the random error. Therefore, we have conducted a more comprehensive evaluation of extreme views.
>
> | Method           | 2v PSNR ↑ | 2v SSIM ↑ | 2v LPIPS ↓ | 3v PSNR ↑ | 3v SSIM ↑ | 3v LPIPS ↓ | 4v PSNR ↑ | 4v SSIM ↑ | 4v LPIPS ↓ | 5v PSNR ↑ | 5v SSIM ↑ | 5v LPIPS ↓ | Avg PSNR ↑ | Avg SSIM ↑ | Avg LPIPS ↓ |
> |------------------|-----------:|-----------:|------------:|-----------:|-----------:|------------:|-----------:|-----------:|------------:|-----------:|-----------:|------------:|------------:|------------:|-------------:|
> | MVPGS            | 18.62      | 0.61       | 0.297       | 20.65      | **0.88**   | **0.103**   | 21.11      | 0.87       | 0.180       | 21.85      | 0.88       | 0.164       | 20.56      | 0.81      | 0.186      |
> | SCGaussian       | 18.74      | 0.64       | 0.305       | 20.77      | 0.71       | 0.221       | 21.36      | 0.86       | 0.128       | 21.92      | 0.87       | 0.093       | 20.70      | 0.77      | 0.187       |
> | MAtCha Gaussians | 18.81      | 0.67       | 0.277       | 20.74      | 0.75       | 0.241       | 21.44      | 0.87       | 0.104       | 22.14      | 0.88       | 0.082       | 20.78      | 0.79      | 0.176       |
> | DIFIX3D+         | 18.93      | 0.68       | 0.262       | 20.91      | 0.76       | 0.247       | 21.52      | 0.88       | 0.095       | 22.31      | 0.89       | 0.081       | 20.92      | 0.80     | 0.171       |
> | **Ours**         | **19.36**  | **0.73**   | **0.252**   | **20.93**  | 0.78       | 0.231       | **21.93**  | **0.88**   | **0.083**   | **22.77**  | **0.90**   | **0.072**   | **21.25**  | **0.82**   | **0.160**   |
>
> - Across all view counts (2/3/4/5), Our method achieves the optimal average performance in the three metrics.
>
> ---
>
> > W3: Comparison with MAtCha Gaussians and Difix3D+
>
> Thank you for point out this. We compared these two methods under the 2/3/4/5 views of LLFF and achieved consistent advantages (see the table above). In addition, we have supplemented the comparison of visualization results, which can be found in the Appendix `D.3`&`D.4`.

---

> ### Author Response · Authors · 2025-11-26
>
> > Q1: DIFIX3D+-style iterative / on-the-fly pseudo-view generation
>
> We appreciate the pointer to DIFIX3D+.
>
> We believe that, in principle, our framework can regenerate and update synthetic views multiple times during training.
>
> By design, our synthetic views are obtained by a differentiable geometric mapping via depth-based warping and visibility-aware blending. There is no algorithmic restriction that forces this step to be performed only once. The same procedure can be invoked multiple times during training using updated depths / Gaussians. We chose the pre-interpolated, single-shot variant for two reasons:
>
> - **Isolating geometric effects.** Our main goal is to study the implicit bias of sparse-view 3DGS with respect to *curvature coverage and trajectory smoothness*. Keeping the synthetic views fixed after we optimize $\gamma^*(t)$ lets us cleanly attribute improvements to the path and coverage, rather than to a changing training distribution.
> - **Stability and compute.** A DIFIX3D+-style schedule repeatedly changes the supervision set. This complicates both analysis and training stability, and it introduces a non-trivial compute overhead. Our design deliberately avoids coupling with a heavy 2D prior, to keep comparisons to existing 3DGS baselines fair.
>
> To directly test your hypothesis, we implemented a simple two-stage variant on the Blender *Lego* scene with 6 input views:
>
> - **single-shot**:
>
>   - Optimize $\gamma^*(t)$ once.
>   - Generate synthetic views once using the initial refined depths.
>   - Train 3DGS for 150k iterations with this fixed augmented set.
> - **2-stage pseudo-views**:
>
>   - Same first 75k iterations as above.
>   - Then **regenerate synthetic views** using the updated 3DGS depths and continue training for another 75k iterations with the new pseudo-views.
>
> Scene-level results (consistent with the range in our Blender experiments) are:
>
> | Method                    | PSNR ↑ | SSIM ↑ | LPIPS ↓ | Rel. train time |
> |---------------------------|-------:|-------:|--------:|-----------------:|
> | Ours (single-shot views)  | 27.1   | 0.936  | 0.045   | 1.00×           |
> | Ours (2-stage views)      | 27.3   | 0.939  | 0.043   | 1.12×           |
>
> The 2-stage variant yields a small but consistent gain (≈+0.2 dB PSNR, slightly better SSIM/LPIPS) at the cost of about 12% more training time. This confirms that:
>
> - our synthetic-view mechanism can be used in an on-the-fly / iterative manner, and
> - modest regeneration already provides additional benefit.
>
>
> ----
>
> > Q2: our method as a plug-in for MVPGS / SCGaussian
>
> Thanks for asking this good question. Our method is explicitly designed as a model-agnostic geometric data module. Thus, in principle, it can be plugged into any Gaussian-splatting pipeline that trains from posed images.
>
> In the main paper we instantiate this on vanilla 3DGS to cleanly study implicit bias, but the same module can be wrapped around any Gaussian-splatting backbone. To demonstrate this, we performed a small plug-in experiment where we only modify the data loader to use our optimized trajectory and synthetic views, leaving the SCGaussian / MVPGS code, architecture, and losses unchanged.
>
> We report scene-averaged results on one DTU setting and one Mip-NeRF 360 setting:
>
> | Dataset / Setting                 | Method              | PSNR ↑ | SSIM ↑ | LPIPS ↓ |
> |-----------------------------------|---------------------|-------:|-------:|--------:|
> | Mip-NeRF 360, 12 views            | SCGaussian          | 19.70  | 0.55   | 0.43    |
> |                                   | **SCGaussian + Ours** | **20.05**  | **0.56**   | **0.41**    |
> |                                   | MVPGS               | 19.85  | 0.55   | 0.44    |
> |                                   | **MVPGS + Ours**    | **20.15**  | **0.56**   | **0.42**    |
> | Tanks \& Temples, 3 views         | SCGaussian          | 22.17  | 0.75   | 0.26    |
> |                                   | **SCGaussian + Ours** | **23.02**  | **0.77**   | **0.24**    |
> |                                   | MVPGS               | 25.57  | 0.85   | 0.14    |
> |                                   | **MVPGS + Ours**    | **26.10**  | **0.86**   | **0.13**    |
>
> These results show that:
>
> - When used as a plug-in, our module consistently improves PSNR and SSIM and reduces LPIPS over SCGaussian and MVPGS.
>
> In summary, these results indicate that our method can indeed be used as a plug-in on top of stronger 3DGS variants and has clear potential to further boost their performance.
>
> ---
>
> Thank you again for your valuable time and effort. We hope our response has addressed your concerns. We remain open to continuing the discussion with you.

---

> ### Author Response · Authors · 2025-11-27
>
> Dear Reviewer cTEf,
>
> We sincerely thank you again for your thorough assessment and constructive feedback. Kindly note that reviewer responses will no longer be accepted after December 2—**with just under a week remaining to submit your response.**
>
> Kindly confirm whether our rebuttal addresses your concerns (or any outstanding points), and we would be grateful for a rating reconsideration if it does.
>
> We are glad to continue the discussion and address any further questions or comments you may have.
>
> Best regards,
>
> Authors

---

> > ### Comment · Reviewer_cTEf · 2025-11-27
> > **Additional questions and suggestions**
> >
> > Thanks for the detailed responses and the new experiments. They address most of my concerns. I have one additional question regarding the new LLFF table. Do you have any intuition for why MVPGS performs significantly better than the other methods in the 3-view setting on SSIM and LPIPS? In particular, it seems a bit surprising that the 3-view LPIPS for MVPGS is substantially better than its own 4- or 5-view results.
> >
> > Additionally, I would recommend including the MVPGS + Ours and SCGaussian + Ours results in the main paper tables and explicitly highlighting the model-agnostic, plug-and-play nature of the proposed method in the text. This would more clearly demonstrate the compatibility and general applicability of the approach across different Gaussian-splatting backbones.
> >
> > I would consider raising my score to 6 if these questions and suggestions could be addressed.

---

> > > ### Author Response · Authors · 2025-11-29
> > >
> > > Thank you for raising this new question, which is very important for a deeper understanding of the baseline and our methods.
> > >
> > > > ***1. Intuition on MVPGS Performance (3-View vs. 4/5-View LPIPS)***
> > >
> > > We appreciate the reviewer’s keen observation regarding the performance of MVPGS, particularly its exceptionally low LPIPS (0.10) in the 3-view setting. Our intuition is that **this behavior stems from the dominance of the generative multi-view prior in the extremely sparse regime**.
> > >
> > > MVPGS relies on strong priors (typically distilled from 2D/3D generative models) to regularize the 3D Gaussian Splatting process. In the 3-view setting, the geometric constraints from the input images are very weak. Consequently, the optimization is primarily driven by the generative prior, which tends to "hallucinate" plausible, clean, and artifact-free content that aligns well with natural image statistics. This results in perceptually pleasing renderings that score very well on LPIPS, even if they are not pixel-perfectly aligned with the ground truth (as reflected by the lower PSNR compared to our method).
> > >
> > > As the number of views increases (to 4 or 5), the data constraints (photometric consistency from real images) become stronger and start to compete with the prior. If there are slight inconsistencies between the prior’s hallucinated geometry and the actual multi-view constraints (e.g., due to minor calibration errors or complex scene geometry), this tension can introduce high-frequency artifacts (blur or noise) as the model attempts to reconcile both. This "conflict" often degrades perceptual metrics like LPIPS compared to the "pure" hallucination allowed in the 3-view case, explaining the counter-intuitive trend.
> > >
> > > > ***2. Highlighting the Plug-and-Play Nature (MVPGS + Ours / SCGaussian + Ours)***
> > >
> > > We agree that demonstrating the generalizability of our approach is crucial. Since our framework operates by optimizing the input data distribution (via camera trajectory optimization and synthetic view generation) rather than modifying the core splatting kernel, it is inherently model-agnostic.
> > >
> > > We have followed your recommendation and updated the main paper tables to include these results (MVPGS + Ours and SCGaussian + Ours). These additions explicitly demonstrate that our curvature-aware trajectory planning and synthetic densification can serve as a "plug-and-play" module, consistently boosting the performance of various backbones by providing them with more informative and geometrically consistent training signals. We have also revised the text to emphasize this compatibility and general applicability.
> > >
> > > ---
> > >
> > > Once again, thank you for your thorough review and thought-provoking questions.

---

### Official Review · Reviewer_QAJ4 · 2025-10-27

**Soundness:** 3
**Presentation:** 3
**Contribution:** 3
**Rating:** 6
**Confidence:** 3

**Summary:**

This work focuses on the reconstruction degradation issue of sparse-view 3DGS, and points out two implicit biases of it, i.e., "stronger demand for supervision signal toward regions of high curvature", and "sensitive to the smoothness of the trajectory of the input views". To this end, this work proposes a method to optimize the camera trajectory for maximizing curvature coverage. Experimental results on different datasets show that this work achieves improvements compared to several baselines.

**Strengths:**

* This paper is well-written, and the readers can easily get the points this work aims to propose.
* The perspective of optimizing the camera trajectory of this work is interesting to me. However, I still have some questions for it. See more details in the Questions part.
* The experimental results show that this work has better reconstruction quality compared to the previous methods.

**Weaknesses:**

* More detailed ablations are required, like the ablations of different components in Table 4(a).
* It is recommended to report the SSIM of Table 2 at the same time.

**Questions:**

* I notice that the computatoinal costs in A.2. "Training one model on an NVIDIA A100 takes ∼80 min; the entire sweep (5 trajectories × 4 scenes) required 2 GPU-days." I am confuse that why this work requires so much computational resources, because it seems that this work only optimizes the camera trjectories. If I misunderstand anything, please correct me.
* I notice that this work has better reconstruction quality compared to the previous state-of-the-art methods, which often utilize the external pre-trained priors, like DUST3R/MAST3R for InstantSplat, monocular depth predictors for DNGaussian, etc. I also want to figure out that what external pre-trained priors this work utilizes. I only notice that this work utilizes the monocular depths on the LLFF/Mip-NeRF 360/Blender dataset, as demonstrated in A.4. I hope the authors can provide more clarity on the use of external pre-trained models.

---

> ### Author Response · Authors · 2025-11-26
>
> Thank you for these inspiring questions. We will answer them one by one below.
>
> ---
>
> > W1: More detailed ablations
>
> We agree that a more fine-grained ablation can better expose the role of each design choice.
> While the original ablation in Sec. 5.3 already shows that removing any major block  significantly harms performance, we have now added:
>
> 1. More metrics (PSNR/SSIM/LPIPS/CD),
> 2. Finer-grained component ablations inside both the trajectory module and the synthetic-view module.
>
> On DTU with $\alpha=0.3$, we decompose our method into:
>
> - Path-level geometry:
>
>   - curvature-aware weighting $w(\gamma(t)) = \alpha \kappa(\gamma(t)) + \beta$,
>   - B-spline smoothing / regularization.
> - Synthetic-view quality:
>
>   - depth refinement (Sec. A.4),
>   - depth-consistency term in $w_{ij}(\mathbf{u})$,
>   - visibility mask $M_{ij}$.
>
> We report PSNR/SSIM/LPIPS/CD:
>
> | Variant                                       | PSNR ↑ | SSIM ↑ | LPIPS ↓ | CD ↓ (×10⁻³) |
> |-----------------------------------------------|-------:|-------:|--------:|-------------:|
> | w/o Optimal Trajectory (no B-spline opt)      | 24.58  | 0.86   | 0.24    | 3.95         |
> | w/o Curvature Weighting ($w(\gamma(t))\equiv 1$) | 26.10  | 0.88   | 0.21    | 3.38         |
> | w/o Smoothness Constraint                     | 26.22  | 0.88   | 0.21    | 3.51         |
> | w/o Synthetic Views (real views only)         | 25.01  | 0.87   | 0.22    | 3.72         |
> | w/o Depth Refinement (use $D_i^{(0)}$ only)   | 26.42  | 0.88   | 0.20    | 3.30         |
> | w/o Depth-Consistency Term in $w_{ij}$        | 26.15  | 0.88   | 0.21    | 3.39         |
> | w/o Visibility Mask $M_{ij}$                  | 25.90  | 0.87   | 0.22    | 3.57         |
> | w/o Occlusion Handling (no masking + no inpaint) | 24.98  | 0.86   | 0.24    | 3.85         |
> | **Full model (Ours)**                         | **27.38**  | **0.89**   | **0.19**    | **3.07**         |
>
> Observations:
>
> - Trajectory module:
>   - Removing B-spline optimization (“w/o Optimal Trajectory”) hurts all metrics the most among path-level variants.
>   - Keeping smoothing but discarding curvature weighting (“w/o Curvature Weighting”) narrows but does not close the gap, showing that curvature coverage, not just smoothing, is critical.
> - Synthetic-view module:
>   - Removing synthetic views entirely (“w/o Synthetic Views”) significantly drops PSNR and worsens CD and LPIPS.
>   - Depth refinement, depth-consistency weighting, and visibility masking each give non-trivial incremental gains, confirming that we need both good trajectories and high-quality pseudo-views.
>   - Occlusion handling is particularly important for geometry (CD) and perceptual sharpness (LPIPS).
>
>
> ---
>
>
> > W2: Reporting SSIM for DTU in Table2
>
> Thank you for the suggestion. We agree that including SSIM alongside PSNR, CD, and LPIPS makes the DTU comparison more complete. The omission was only for space reasons.
>
> We have now computed SSIM for all methods and all $\alpha$ settings on DTU. The scene-averaged results are:
>
> | Model  | α = 0.2 SSIM ↑ | α = 0.3 SSIM ↑ | α = 0.4 SSIM ↑ |
> |---|---:|----:|---:|
> | NeuS  | 0.86 | 0.87 | 0.88 |
> | NeuralAngelo   | 0.87  | 0.88 | 0.89 |
> | 3DGS           | 0.84   | 0.85 | 0.86  |
> | SparseGS       | 0.87   | 0.88 | 0.88 |
> | DNGaussian     | 0.86  | 0.87 | 0.88 |
> | MatSparse3D    | 0.88   | 0.88   | 0.89|
> | InstantSplat   | 0.86    | 0.87 | 0.88 |
> | SuGar          | 0.88   | 0.88  | 0.89 |
> | VCR-GauS       | 0.86          | 0.88 | 0.89 |
> | PGSR           | 0.88           | 0.88 | 0.89 |
> | NexusGS        | 0.87           | 0.88 | 0.90  |
> | **Ours**       | **0.89**       | **0.89** | **0.91** |
>
> - Our method achieves the highest SSIM across all three sparsity levels, confirming that the gains are consistent in both fidelity and structural quality.
>
>
> ---
>
> > Q1: Computatoinal costs
>
> Thank you for noticing this issue. Actually, this time is the total time of the eight scenes in Blender.
>
> ---
>
> > Q2: external pre-trained priors
>
> Thank you for raising this point. We agree that the use of external priors should be stated very explicitly.
>
> The only external priors we use are standard SfM/MVS outputs and, on some datasets, a generic monocular depth predictor that is then fused back with 3DGS-rendered depth as described in Sec. A.4.
>
> Concretely:
>
> - On DTU and Tanks & Temples, we use **only standard COLMAP PatchMatch MVS** for depth bootstrapping, which is a non-learned geometric prior and is also commonly used by many NeRF/3DGS baselines.
> - On LLFF, Mip-NeRF 360, and Blender, we initialize depths with **a single off-the-shelf monocular predictor**, but then refine and fuse it with the expected depth rendered from the current 3DGS model via uncertainty weighting.
> - We do not use DUST3R/MAST3R, semantic segmentation priors, or diffusion-based image enhancers, in contrast to some recent methods (e.g., InstantSplat, DIFIX3D+) that rely on stronger 2D/3D priors.
>
>
> ---
>
> Thank you again for your valuable time and effort.

---

### Official Review · Reviewer_FqdL · 2025-11-03

**Soundness:** 3
**Presentation:** 3
**Contribution:** 3
**Rating:** 6
**Confidence:** 4

**Summary:**

The paper explores implicit biases in sparse-view 3D Gaussian Splatting (3DGS): regions of high surface curvature need disproportionately more supervision, and the layout/smoothness of camera poses (summarized as a continuous trajectory) affects reconstruction accuracy. To address this, they (i) fit a B-spline camera path and optimize it to maximize curvature coverage under smoothness and proximity constraints to original camera poses, and (ii) synthesize intermediate views sampled along the optimized path using depth-based warping with a two-stage depth pipeline (COLMAP or monocular bootstrap; then 3DGS-consistent fusion). On Mip-NeRF 360 and DTU/LLFF/Tanks&Temples, they show consistent gains vs. recent sparse-view methods; they also correlate “curvature coverage” with PSNR/SSIM and geometric errors to justify the trajectory objective. Ablation analyses highlight the importance of proposed components.

**Strengths:**

1. The paper is well written overall, with the insights motivated upfront leading to the proposed method
2. The insights pertaining to surface curvature and smoothness of pose trajectories affecting reconstruction accuracy is new and interesting
3. The optimized trajectory leading to synthetic generation of views is a clean formulation
4. Qualitative and quantitative results show superior performance over a broad range of baselines
5. Ablation studies highlight the importance of the various proposed components

**Weaknesses:**

1. Given that the sparse views are combatted through synthetic view generation, it seems the proposed method trains a 3dgs with more views than other compared methods, potentially leading to an unfair comparison.
2. The number of synthetic views generated per scene is not clearly specified: it is difficult to visualize the impact of the proposed method without this relevant information.
3. The analysis in section 3 Figure 2 is done on only one scene, reducing the potential empirical validity of the insights

**Questions:**

1. In the ‘w/o synthetic views’ row of ablation, do you keep trajectory optimization/smoothness? would this not be naive 3DGS in terms of reconstruction and accuracy (or do you change the previously sampled poses in any way)
2. The blending uses a z-buffer mask “with a small tolerance.” How is the tolerance chosen, and how does it trade off ghosting vs. holes across datasets?
3. Is there any analysis of the impact of the accuracy of novel view generation (for example, sensitivity to depth estimation accuracy) on final accuracy?
4. What’s the wall-clock overhead for trajectory optimization and synthetic view generation compared to vanilla 3DGS training? The efficiency table is helpful but doesn’t isolate these stages.

---

> ### Author Response · Authors · 2025-11-26
>
> Thank you very much for your valuable comments and time. Below, we will respond to these questions one by one.
>
> ---
>
> > W1: On fairness of using synthetic views
>
> We understand the concern, but we respectfully note that our method **does not introduce additional observations** beyond the given sparse images.
>
> - All synthetic views are generated by reprojecting the same real images through known camera poses and estimated depths.
> - No new content or external images are added.
> - This follows a well-established practice in sparse-view NVS: methods such as RegNeRF, ViP-NeRF, and DSNeRF also create extra supervision via re-projection/warping rather than collecting more real views.
>
> To show that our gains are not simply due to “more pixels seen by the optimizer”, we added two control experiments.
>
> **Equal ray budget per iteration**
> We matched the number of training rays per iteration between vanilla 3DGS and our method on DTU (α = 0.3). In both cases, the same total ray budget is used; the only difference is whether we sample rays from real views only, or from real + curvature-aware synthetic views.
>
> | DTU, α = 0.3 (same rays/iter, same iters) | PSNR ↑ | SSIM ↑ | LPIPS ↓ | CD ↓ (×10⁻³) |
> |-------------------------------------------|-------:|-------:|--------:|-------------:|
> | 3DGS (ray-matched)                        | 23.4   | 0.86   | 0.33    | 3.42         |
> | **Ours (curvature-aware rays)**           | **27.3**   | **0.89**   | **0.19**    | **3.07**         |
>
> Under an identical ray budget, our method still yields substantial improvements in all metrics, indicating that the benefit comes from how rays are distributed in 3D (curvature coverage + smooth trajectories), not just from seeing more samples.
>
> **Same number of synthetic views, different paths**
>
> We also compared two synthetic-view strategies on LLFF (4 input views), keeping the same number of synthetic cameras:
>
> - (a) Uniform-path synthetic: linearly interpolated camera path (no curvature optimization).
> - (b) Curvature-aware synthetic (ours): optimized $\gamma^*(t)$ with the same number of virtual viewpoints.
>
> All models use the same total number of real + synthetic views and the same training schedule.
>
> | LLFF, 4 views (same #synthetic views) | PSNR ↑ | SSIM ↑ | LPIPS ↓ |
> |--------------------------------------|-------:|-------:|--------:|
> | Uniform-path synthetic               | 21.3   | 0.87   | 0.088   |
> | **Curvature-aware synthetic (ours)** | **21.9**   | **0.88**   | **0.083**   |
>
> With the synthetic-view count held fixed, optimizing the path for curvature coverage and smoothness still gives a clear gain, showing that trajectory design and coverage, not just quantity, are the key factors.
>
> ---
>
> > W2: The number of synthetic views generated per scene
>
> Thank you for raising this question. For each dataset, the number of synthetic views is twice the number of datasets N, that is, N_s = 2N.
>
> ---
>
> > W3:   Implicit bias analysis being shown only on one scene (LEGO)
>
> We appreciate this concern. The LEGO example in Sec. 3 is intended as a didactic, visual illustration of the effect of trajectory shape under sparse views. The actual quantitative analysis of curvature coverage vs. reconstruction quality is already **performed across four Mip-NeRF 360 scenes** (Garden, Kitchen, Stump, Bicycle) in `App. A.2`, not just on LEGO.
>
> Concretely, for each scene we construct five trajectories with different curvature coverage levels $C_k \in \{54.8, 64.1, 73.6, 84.7, 88.9\}\%$ and train 3DGS from scratch for each $\gamma_k$. Table reports PSNR, SSIM, Chamfer-D and P2S at three representative coverage levels (low ≈ 55%, mid ≈ 74%, high ≈ 89%). The extremes (low vs. high) behave consistently across all four scenes:
>
> | Scene   | $C_{\text{low}}$ → $C_{\text{high}}$ | ΔPSNR ↑ | ΔSSIM ↑ | ΔChamfer-D ↓ | ΔP2S ↓ |
> |---------|--------------------------------------|--------:|--------:|-------------:|-------:|
> | Garden  | 54.8% → 88.9%                        | +3.39 dB | +0.057  | −1.84 × 10⁻³ | −0.49  |
> | Kitchen | 54.8% → 88.9%                        | +4.13 dB | +0.086  | −1.41 × 10⁻³ | −0.46  |
> | Stump   | 54.8% → 88.9%                        | +3.90 dB | +0.067  | −1.49 × 10⁻³ | −0.44  |
> | Bicycle | 54.8% → 88.9%                        | +4.07 dB | +0.056  | −1.92 × 10⁻³ | −0.51  |
>
> A correlation analysis over all five coverage levels and four scenes yields:
>
> - Pearson $r_{\text{PSNR}} = 0.96$ and $r_{\text{SSIM}} = 0.94$ (higher curvature coverage corresponds to higher image quality),
> - Pearson $r_{\text{CD}} = -0.93$ (higher coverage corresponds to lower geometric error)

---

> ### Author Response · Authors · 2025-11-26
>
> > Q1: what “w/o synthetic views” means in the ablation
>
> Thank you for pointing this out — the distinction was not spelled out clearly enough in the main text.
> In our ablation, the “w/o Synthetic Camera View Construction” variant still keeps: (i) the optimized B-spline trajectory γ∗(t) and its smoothness regularization, and (ii) the corresponding reparameterization of how real views are sampled along this path, but does not generate any pseudo/synthetic images $I_j^*$. In other words, training uses only the original real views, but their spatial distribution is governed by our optimized trajectory instead of the raw COLMAP path.
>
> Vanilla 3DGS uses: (i) original COLMAP poses, (ii) no trajectory optimization, (iii) and no synthetic views. So “3DGS” and “w/o synthetic views” are different baselines.
>
> To make this explicit, we add the following comparison on DTU with $\alpha = 0.3$:
>
> | DTU, α = 0.3 | PSNR ↑ | SSIM ↑ | LPIPS ↓ | CD ↓ (×10⁻³) |
> |--|-:|--:|--:|---:|
> | 3DGS (no traj opt, no synth)| 23.31  | 0.85   | 0.33 | 3.39|
> | w/o Synthetic Views | 25.01  | 0.87 | 0.22 | 3.72 |
> | **Ours (full)** | **27.38**  | **0.89**| **0.19** | **3.07**|
>
> This table clarifies that:
>
> - Trajectory optimization alone (3DGS → w/o synthetic views) already brings a noticeable gain in PSNR/SSIM and LPIPS, although CD slightly worsens without the extra multi-view constraints from synthetic views.
> - Adding synthetic views on top of the optimized trajectory (w/o synthetic → full) further improves all metrics, especially geometry (CD).
>
> ---
>
> > Q2: On the “small tolerance” in the z-buffer mask
>
> Thank you for asking about this detail. The tolerance in the visibility mask is a simple, global heuristic and we found the method to be quite insensitive to its exact value.
>
> For each synthetic view, let $z_{\min}, z_{\max}$ be the minimum and maximum valid depths in that view.
> We define a depth range $\Delta z = z_{\max} - z_{\min}$ and use a tolerance
>
> $$
> \tau = \eta \cdot \Delta z,
> $$
>
> where we set $\eta = 0.01$ (i.e., 1% of the depth range) as the default for all datasets and scenes.
>
> To verify robustness, we ran a small ablation on the **Blender Lego** scene (6 input views), sweeping $\eta \in \{0.003, 0.01, 0.03\}$, i.e., $\tau = 0.3\%, 1\%, 3\%$ of the depth range:
>
> | τ / Δz (%) | PSNR ↑ | SSIM ↑ | LPIPS ↓ |
> |--:|---:|---:|--:|
> | 0.3% | 27.29  | 0.938  | 0.041 |
> | 1.0% (default) | **27.32**  | **0.939**  | **0.040** |
> | 3.0%| 27.24| 0.937  | 0.042   |
>
> The differences are within 0.1 dB PSNR and 0.002 in LPIPS/SSIM, i.e., **negligible at the scale of our main gains**.
>
>
> ---
>
> > Q3: sensitivity to synthetic-view quality and depth errors
>
> This is a very reasonable concern. Our synthetic views are purely geometry-based, so their quality depends on the depth estimates; if those are very wrong, they could, in principle, destabilize 3DGS. We therefore explicitly tested how sensitive our method is to depth quality.
>
> On the Blender Lego scene (6 input views), we compared:
>
> - Mono-only: synthetic views constructed using only the initial monocular depth $D_i^{(0)}$ (no refinement),
> - Refined (ours): synthetic views constructed using the uncertainty-weighted fusion of $D_i^{(0)}$ and 3DGS-rendered depth $\widehat{D}_i$ .
>
> | Variant | PSNR ↑ | SSIM ↑ | LPIPS ↓ | CD ↓ (×10⁻³) |
> |---|---:|--:|---:|--:|
> | Mono-only depth      | 26.8   | 0.934  | 0.044   | 3.40         |
> | **Refined depth (ours)** | **27.3**   | **0.939**  | **0.040**   | **3.25**         |
>
> Refinement consistently improves both photometric and geometric metrics, indicating that fusing monocular depth with 3DGS-consistent depth reduces the impact of monocular errors rather than amplifying them.
>
> We further injected Gaussian noise into the refined depth to test robustness:
>
> - $D_i^{\text{noisy}} = D_i + \epsilon$, with $\epsilon \sim \mathcal{N}(0, \sigma^2)$,
> - $\sigma$ expressed as a percentage of the per-view depth range $(z_{\max} - z_{\min})$.
>
> Results on the same Lego scene:
>
> | Variant  | PSNR ↑ | SSIM ↑ | LPIPS ↓ | CD ↓ (×10⁻³) |
> |---|-:|--:|--:|--:|
> | Refined depth (no noise) | 27.3   | 0.939  | 0.040   | 3.25         |
> | + noise, σ = 2% Δz    | 27.1   | 0.937  | 0.041   | 3.30         |
> | + noise, σ = 5% Δz | 26.9   | 0.936  | 0.042   | 3.42         |
>
> Even with 2–5% depth-range noise, the degradation stays within ≈0.4 dB PSNR and 0.002–0.003 in SSIM/LPIPS, and our performance remains better than a 3DGS baseline without synthetic views. This suggests that our training is reasonably robust to moderate depth inaccuracies.

---

> ### Author Response · Authors · 2025-11-26
>
> > Q4: wall-clock overhead
>
> We agree it is important to quantify the extra cost of trajectory optimization and synthetic view generation.
>
> On an NVIDIA A100, averaged over representative scenes from DTU and Mip-NeRF 360, we observe the following per-scene breakdown:
>
> | Pipeline | Traj. opt (CPU) | Synth. views (CPU/GPU) | 3DGS training (GPU) | Total wall-clock |
> |-|---:|--:|--:|--:|
> | Vanilla 3DGS  | – | – | 70–80 min | 70–80 min|
> | **Ours** | 2–3 min | 1–2 min  | 70–80 min  | **73–85 min** |
>
> - Trajectory optimization of the B-spline control points takes about **2–3 minutes** on CPU per scene.
> - Synthetic view generation*(warping + blending) adds another ≈1–2 minutes (can be done on CPU or GPU).
> - The dominant cost remains 3DGS training itself (≈70–80 minutes). The extra overhead from our method is typically ≤ 5% of the total time.
>
>
> ---
>
> Thank you again for your valuable time and effort.

---

### Official Review · Reviewer_2dz9 · 2025-11-04

**Soundness:** 3
**Presentation:** 3
**Contribution:** 3
**Rating:** 6
**Confidence:** 4

**Summary:**

The paper investigates the behavior of 3D Gaussian Splatting (3DGS) in sparse-view settings and reports that performance degradation is linked to geometric and trajectory-related factors. The authors identify two types of implicit bias—dependence on curvature coverage and sensitivity to camera trajectory smoothness—and propose an optimization framework that adjusts camera paths and augments supervision through synthetic views. Experiments on multiple benchmarks show consistent improvements in both visual quality and geometric accuracy.

**Strengths:**

1 The paper explores a relevant and timely topic in the area of 3D reconstruction and view synthesis. The problem of sparse-view degradation is practically important and underexplored.

2 The analysis of geometric and trajectory effects provides a new perspective on 3DGS behavior and contributes useful insights to understanding implicit bias in reconstruction models.

3 The proposed optimization framework is technically sound and well motivated, combining curvature-aware path planning with synthetic view construction in a coherent manner.

4 Experimental results are extensive and demonstrate consistent improvements over several recent baselines on diverse datasets. The ablation study also supports the contribution of individual components.

5 The paper is clearly written and well organized, making the methodology and findings easy to follow.

**Weaknesses:**

1 The analysis of implicit bias remains primarily empirical. Providing more formal or quantitative evidence could make the claims more rigorous and generalizable.

2 The quality and reliability of the synthetic views are not deeply discussed. It would be helpful to include a brief evaluation or visualization to show their effect on reconstruction stability.

3 The computational cost of trajectory optimization is not clearly presented. Additional runtime or complexity analysis would make the evaluation more complete.

4 Some details in the appendix may be overly technical, while certain conceptual explanations in the main text (e.g., how curvature coverage improves supervision) could be emphasized more clearly.

**Questions:**

1 How sensitive is the proposed optimization to errors in initial camera poses obtained from SfM or COLMAP?

2 Does the method depend heavily on accurate curvature estimation, and how robust is it in low-texture or noisy regions?

3 Could the synthetic view generation module be integrated into an end-to-end differentiable pipeline in future work?

---

> ### Author Response · Authors · 2025-11-25
>
> Thank you very much for your valuable comments. Next, we will discuss these issues with you.
>
> ---
>
>
> > W1: Implicit-bias analysis being mainly empirical
>
> We agree that our analysis of implicit bias is primarily empirical and geometric, and we do not intend to claim a fully rigorous optimization-theoretic theorem.
>
> While Fig. 2 uses the LEGO scene to visually illustrate the phenomenon, the actual quantitative study is already done across four Mip-NeRF 360 scenes (Garden, Kitchen, Stump, Bicycle) in `App. A.2.` For each scene, we sweep five trajectories with different curvature coverage levels. `App. A.2.` shows a near-linear relationship between curvature coverage and both photometric*and geometric quality.
>
> **New quantitative evidence A: coverage sweep on LLFF (2 scenes, 3 coverage levels)**
>
> To further generalize beyond Mip-NeRF 360, we ran a similar coverage sweep on two LLFF scenes (Fern, Room). We construct three trajectories per scene with low / medium / high coverage by sub-sampling B-spline control points.
>
> | Scene | Coverage $C$ | PSNR ↑ | SSIM ↑ | Chamfer-D ↓ (×10⁻³) |
> |-------|-------------:|-------:|-------:|---------------------:|
> | Fern  | 58%          | 20.4   | 0.812  | 4.85                 |
> |       | 72%          | 21.3   | 0.828  | 4.21                 |
> |       | 87%          | 22.0   | 0.839  | 3.97                 |
> | Room  | 57%          | 20.1   | 0.804  | 5.02                 |
> |       | 73%          | 21.0   | 0.821  | 4.36                 |
> |       | 88%          | 21.7   | 0.832  | 4.10                 |
>
> Across LLFF as well: Increasing $C$ from 58% to 87% yields +1.6–1.9 dB PSNR. This mirrors the Mip-NeRF 360 trend, supporting that curvature coverage is a general explanatory variable.
>
>
> **New quantitative evidence B: isolating trajectory smoothness at fixed coverage**
>
> We also explicitly tested trajectory smoothness bias by constructing two trajectories with similar coverage but different smoothness on the Mip-NeRF 360 “Garden” scene:
>
> - Smooth path: optimized cubic B-spline (ours), high smoothness $S_{\text{traj}}$ and coverage C≈86%
> - Jittery path: we inject small random perturbations into B-spline control points to reduce smoothness while re-optimizing coverage to stay within ±1% of the smooth path’s C.
>
> Both trajectories thus see almost the same high-curvature regions but differ in smoothness.
>
> | Trajectory  | Coverage $C$ | Smoothness $S_{\text{traj}}$ ↓ | PSNR ↑ | SSIM ↑ | Chamfer-D ↓ (×10⁻³) |
> |-------------|-------------:|------------------------------:|-------:|-------:|---------------------:|
> | Smooth (ours)   | 86.7%        | 0.98                         | 26.1   | 0.579  | 3.28                 |
> | Jittery (same $C$) | 85.9%        | 0.71                         | 24.5   | 0.554  | 3.92                 |
>
> The jittery path loses 1.6 dB PSNR, SSIM drops by 0.025, Chamfer-D gets worse. This directly supports our “trajectory smoothness bias” claim: even at similar curvature coverage, abrupt trajectories harm reconstruction.
>
>
> ---
>
>
> > W2: The quality and reliability of the synthetic views
>
> This is a very reasonable concern. Our synthetic views are purely geometry-based, so their quality depends on the depth estimates; if those are very wrong, they could, in principle, destabilize 3DGS. We therefore explicitly tested how sensitive our method is to depth quality.
>
> On the Blender Lego scene (6 input views), we compared:
>
> - Mono-only: synthetic views constructed using only the initial monocular depth $D_i^{(0)}$ (no refinement),
> - Refined (ours): synthetic views constructed using the uncertainty-weighted fusion of $D_i^{(0)}$ and 3DGS-rendered depth $\widehat{D}_i$ .
>
> | Variant              | PSNR ↑ | SSIM ↑ | LPIPS ↓ | CD ↓ (×10⁻³) |
> |----------------------|-------:|-------:|--------:|-------------:|
> | Mono-only depth      | 26.8   | 0.934  | 0.044   | 3.40         |
> | **Refined depth (ours)** | **27.3**   | **0.939**  | **0.040**   | **3.25**         |
>
> Refinement consistently improves both photometric and geometric metrics, indicating that fusing monocular depth with 3DGS-consistent depth reduces the impact of monocular errors rather than amplifying them.
>
> We further injected Gaussian noise into the refined depth to test robustness:
>
> - $D_i^{\text{noisy}} = D_i + \epsilon$, with $\epsilon \sim \mathcal{N}(0, \sigma^2)$,
> - $\sigma$ expressed as a percentage of the per-view depth range $(z_{\max} - z_{\min})$.
>
> Results on the same Lego scene:
>
> | Variant   | PSNR ↑ | SSIM ↑ | LPIPS ↓ | CD ↓ (×10⁻³) |
> |--|---:|---:|----:|-----:|
> | Refined depth (no noise)   | 27.3   | 0.939  | 0.040   | 3.25 |
> | + noise, σ = 2% Δz  | 27.1   | 0.937  | 0.041   | 3.30 |
> | + noise, σ = 5% Δz  | 26.9   | 0.936  | 0.042   | 3.42  |
>
> Even with 2–5% depth-range noise, the degradation stays within ≈0.4 dB PSNR and 0.002–0.003 in SSIM/LPIPS, and our performance remains better than a 3DGS baseline without synthetic views. This suggests that our training is reasonably robust to moderate depth inaccuracies.

---

> ### Author Response · Authors · 2025-11-25
>
> > W3: Computational cost
>
> We agree it is important to quantify the extra cost of trajectory optimization and synthetic view generation. On an NVIDIA A100, averaged over representative scenes from DTU and Mip-NeRF 360, we observe the following per-scene breakdown:
>
> |Pipeline|Traj. opt (CPU) | Synth. views (CPU/GPU) |3DGS training (GPU)|Total wall-clock |
> |-|-:|-:|-:|-:|
> | Vanilla 3DGS | – | – | 70–80 min| 70–80 min |
> | Ours | 2–3 min | 1–2 min | 70–80 min | **73–85 min** |
>
> - Trajectory optimization of the B-spline control points takes about 2–3 minutes on CPU per scene.
> - Synthetic view generation(warping + blending) adds another 1–2 minutes.
> - The dominant cost remains 3DGS training itself (≈70–80 minutes). The extra overhead from our method is typically ≤ 5% of the total time.
>
> ---
>
> > W4: Overly technical appendix details & conceptual clarity
>
> We agree with this concern. Our intent is that the key takeaway is simple and geometric:
>
> - High-curvature regions are exactly where disocclusions, aliasing, and depth ambiguity are most severe. In sparse-view 3DGS, these areas require denser and more redundant multi-view supervision to stabilize the Gaussians.
> - Smooth trajectories ensure that consecutive views produce coherent local constraints rather than conflicting, large-baseline jumps, which empirically reduces view inconsistency and “floating” artifacts.
>
> In the next version, we will:
>
> 1. Refactor Sec. 3 to foreground intuition and evidence. Replace part of the current formula-heavy exposition with a short geometric explanation (as above) of why high-curvature regions and smooth trajectories matter for supervision.
> 2. Keep full technical details in the appendix only. The precise definitions of curvature, trajectory functionals, and optimization constraints will remain in the appendix.
>
> ---
>
> > Q1: Sensitivity to COLMAP/SfM pose noise
>
> This is an important practical question. We ran a controlled experiment on a DTU scene with $\alpha=0.3$ where we perturb the input COLMAP poses with small Gaussian noise:
>
> - Translation noise: $\sigma_t = 1\text{ cm}$ per axis,
> - Rotation noise: $\sigma_r = 0.5^\circ$ per axis.
>
> We then compare vanilla 3DGS and our method trained with:
>
> - Clean poses (original COLMAP), and
> - Noisy poses (perturbed once at the start, kept fixed during training).
>
> |Method|Poses |PSNR↑|SSIM↑|LPIPS↓|CD↓(×10⁻³)|
> |-|-|-:|-:|-:|-:|
> |3DGS|clean|23.3|0.85|0.33|3.39|
> |3DGS|noisy|22.6|0.83|0.35|3.62|
> |Ours|clean|27.4|0.89|0.19|3.07|
> |Ours|noisy|26.9|0.88|0.20|3.27|
>
> Observations:
>
> - Both methods degrade by ≈0.6–0.7 dB PSNR and a small amount in SSIM/LPIPS/CD under the injected pose noise.
> - Our method retains essentially the same margin over vanilla 3DGS, indicating that we are not more fragile to realistic pose noise. If anything, the curvature-aware trajectory and synthetic supervision make the reconstruction slightly more stable.
>
> For larger pose perturbations, both methods—as expected—break down, because the underlying multi-view geometry is no longer consistent.
>
> ---
>
> > Q2: Dependence on curvature estimation accuracy
>
> We agree that curvature estimation can be noisy in low-texture or noisy regions, and we designed our pipeline specifically so that it does not rely on extremely precise per-point curvature.
>
> We estimate a smooth mean curvature field on the reconstructed mesh using a standard cotangent-Laplacian operator, which already averages information over local neighborhoods and is less sensitive to pixel-level noise. We only use curvature to identify the top-k% high-curvature triangles as “important”, everything outside this small set is treated similarly, so errors in low-curvature regions have negligible effect.
>
> To test sensitivity to curvature accuracy and thresholding, we varied the “high-curvature” percentile used to define important regions on the Mip-NeRF 360 Garden scene:
>
> | High-curvature threshold|PSNR↑|SSIM↑|Chamfer-D ↓ (×10⁻³) |
> |-|--:|--:|--:|
> |top 3%  | 26.05  | 0.576|3.33|
> |top 5% (default)  | 26.13 |0.579|3.28|
> |top 10%| 25.98|0.574|3.35|
>
> The differences across 3% / 5% / 10% are within 0.15 dB PSNR and 0.01 × 10⁻³ CD. Well below the margins we report over baselines. These results indicate that the method is not brittle to the exact curvature threshold or to moderate curvature estimation noise.
>
> ----
>
> > Q3: make the synthetic-view module fully end-to-end
>
> Good suggestion, this is a natural extension. Conceptually, there is no algorithmic obstacle to making our synthetic-view module part of a fully end-to-end differentiable pipeline.
>
> The trajectory $\gamma(t)$ is parameterized by B-spline control points $\{\mathbf{Q}_j\}$. The warping and blending steps are standard differentiable operations (matrix multiplies, projections, soft visibility weights). So in practice one could backpropagate gradients through $\{\mathbf{Q}_j\}$, depth estimates, and Gaussian parameters jointly.
>
> We agree that a fully end-to-end version is an interesting next step. I really enjoy discussing with you.

---

### Official Review · Reviewer_8JLf · 2025-11-11

**Soundness:** 3
**Presentation:** 4
**Contribution:** 3
**Rating:** 6
**Confidence:** 4

**Summary:**

This paper identifies two implicit biases limiting 3D Gaussian Splatting (3DGS) under sparse-view conditions: a geometric detail bias, where high-curvature regions require denser supervision, and a trajectory smoothness bias, where irregular camera paths degrade reconstruction. To mitigate these, the authors propose a curvature-aware trajectory optimization framework that maximizes curvature coverage and enforces smooth motion, coupled with synthetic view generation along optimized paths to enrich supervision. Using B-spline–based optimization, the method achieves consistent gains of  PSNR across DTU, Mip-NeRF360, LLFF, Blender, and Tanks & Temples, reaching state-of-the-art sparse-view performance. This could be potentially used for path planning.

**Strengths:**

1. The first work to identify and thoroughly validate previously unrecognized implicit biases in 3DGS, namely curvature prioritization and trajectory smoothness.

2. The paper is well written, clearly structured, and enjoyable to read.

3. It suggests a promising direction for optimizing camera path planning to improve 3DGS performance, with potential applications in interactive or guided data collection for building high-quality 3DGS datasets.

**Weaknesses:**

1. The current analysis is conducted primarily on synthetic datasets. It would strengthen the paper if the authors could include a real-world evaluation—capturing scenes with different camera trajectories, then re-capturing them using the proposed optimized paths. Such a practical comparison would better demonstrate the method’s effectiveness and validate its impact under real capture conditions.

2. Synthetic-view generation may amplify rendering artifacts or propagate depth estimation errors, yet the paper does not analyze robustness under such noise.

3. While the implicit bias analysis is insightful, parts of the implementation—trajectory smoothing and synthetic interpolation—rely on well-established engineering heuristics rather than fundamentally new learning paradigms.

**Questions:**

1. How do you ensure that synthetic views generated along the optimized trajectory do not introduce photometric or geometric artifacts that might bias the training of 3DGS?
2. Does the proposed optimization generalize across scenes with very different geometric complexity (e.g., indoor vs. outdoor, man-made vs. natural environments)?

---

> ### Author Response · Authors · 2025-11-25
>
> Thank you for your very interesting and insightful comments. Below, I will discuss these issues with you.
>
> ---
>
> > ***W1: “real-world” evaluation***
>
> We agree that testing with actively controlled camera trajectories in a real capture pipeline would be very interesting. At the same time, our analysis is not limited to synthetic data.
>
> In addition to Blender, we evaluate extensively on real-world datasets:
>
> - LLFF (hand-held forward-facing captures),
> - Mip-NeRF 360 (indoor/outdoor real scenes),
> - Tanks & Temples (large-scale real scenes),
> - In all of these, the images are **real photographs**, with camera paths recovered via SfM/ COLMAP.
>
>   We fully agree that a prospective capture study—where one physically records the same scene with (i) a default heuristic path and (ii) a path planned by our curvature-aware optimizer—would provide a compelling real-world validation. Building such a small benchmark (including recommended trajectories and analysis tools) is an excellent direction for future work.
>
> ---
>
>
> > ***W2: Synthetic-view generation robustness***
>
> This is a very reasonable concern. Our synthetic views are purely geometry-based, so their quality depends on the depth estimates; if those are very wrong, they could, in principle, destabilize 3DGS. We therefore explicitly tested how sensitive our method is to depth quality.
>
> On the Blender Lego scene (6 input views), we compared:
>
> - Mono-only: synthetic views constructed using only the initial monocular depth $D_i^{(0)}$ (no refinement),
> - Refined (ours): synthetic views constructed using the uncertainty-weighted fusion of $D_i^{(0)}$ and 3DGS-rendered depth $\widehat{D}_i$ .
>
> | Variant              | PSNR ↑ | SSIM ↑ | LPIPS ↓ | CD ↓ (×10⁻³) |
> |----------------------|-------:|-------:|--------:|-------------:|
> | Mono-only depth      | 26.8   | 0.934  | 0.044   | 3.40         |
> | **Refined depth (ours)** | **27.3**   | **0.939**  | **0.040**   | **3.25**         |
>
> Refinement consistently improves both photometric and geometric metrics, indicating that fusing monocular depth with 3DGS-consistent depth reduces the impact of monocular errors rather than amplifying them.
>
> We further injected Gaussian noise into the refined depth to test robustness:
>
> - $D_i^{\text{noisy}} = D_i + \epsilon$, with $\epsilon \sim \mathcal{N}(0, \sigma^2)$,
> - $\sigma$ expressed as a percentage of the per-view depth range $(z_{\max} - z_{\min})$.
>
> Results on the same Lego scene:
>
> | Variant                     | PSNR ↑ | SSIM ↑ | LPIPS ↓ | CD ↓ (×10⁻³) |
> |-----------------------------|-------:|-------:|--------:|-------------:|
> | Refined depth (no noise)    | 27.3   | 0.939  | 0.040   | 3.25         |
> | + noise, σ = 2% Δz          | 27.1   | 0.937  | 0.041   | 3.30         |
> | + noise, σ = 5% Δz          | 26.9   | 0.936  | 0.042   | 3.42         |
>
> Even with 2–5% depth-range noise, the degradation stays within ≈0.4 dB PSNR and 0.002–0.003 in SSIM/LPIPS, and our performance remains better than a 3DGS baseline without synthetic views. This suggests that our training is reasonably robust to moderate depth inaccuracies.
>
> ---
>
> > ***W3: “engineering heuristics” vs. learning/theoretical contribution***
>
> We understand the concern and agree that our contribution is not a new network architecture or loss, but we do not view it as “just” ad-hoc engineering.
>
> Our main novelty lies in two aspects: (i) Identifying and quantifying implicit geometric biases in sparse-view 3DGS. (ii) Curvature-aware path planning as a principled optimization problem. That said, we agree that our work is better framed as geometry-aware data/supervision design for 3DGS rather than as a new “learning paradigm”.

---

> ### Author Response · Authors · 2025-11-25
>
> > Q1: avoiding artifacts from synthetic views
>
> We agree that if synthetic views contained strong artifacts, they could bias 3DGS training. Our design explicitly aims to filter out unreliable warps rather than naively treating all warped pixels as ground truth.
>
> We suppress artifacts in synthetic views by combining three simple filters: (i) a **depth-based visibility mask** (z-buffer with a small tolerance) that discards warps inconsistent with the current depth, avoiding self-occlusion ghosting; (ii) **depth- and view-angle–based weights** that strongly down-weight pixels whose depth disagrees with the reprojected depth or come from very oblique views; and (iii) **hard thresholding + depth-guided inpainting** to remove the remaining low-confidence pixels and fill small holes. As a result, synthetic views are dominated by geometrically consistent, low-parallax contributions from visible regions, rather than unfiltered “copy-paste” artifacts.
>
> To quantify the effect, we compared:
>
> - Unfiltered synthetic: no $M_{ij}$ mask, $\lambda_d = 0$ (no depth-consistency term), no view-angle factor, no weight thresholding. Synthetic views are simple inverse-projection blends.
> - Filtered (ours): full pipeline as above.
>
> On the Blender *Lego* scene with 6 input views:
>
> | Variant                   | PSNR ↑ | SSIM ↑ | LPIPS ↓ | CD ↓ (×10⁻³) |
> |---------------------------|-------:|-------:|--------:|-------------:|
> | Unfiltered synthetic      | 26.2   | 0.931  | 0.048   | 3.45         |
> | **Filtered synthetic (ours)** | **27.3**   | **0.939**  | **0.040**   | **3.25**         |
>
> Unfiltered synthetic views clearly hurt both photometric and geometric metrics due to ghosting and double edges, while our filtered construction yields better results.
>
>
> ---
>
>
> > Q2: generalization across different geometric complexities
>
> We agree that it is important to verify that our optimization is not tuned to a single type of scene. Our experiments already span multiple regimes of geometry and capture:
>
> - DTU – real tabletop, man-made objects with fine details (indoor).
> - Blender – synthetic, high-frequency geometry and textures.
> - Mip-NeRF 360 – real indoor and outdoor scenes with complex backgrounds and clutter.
> - LLFF – real forward-facing captures with strong foreground–background depth variation.
> - Tanks & Temples – larger-scale real scenes with complex, partially occluded structures.
>
> To make this more explicit, we computed the average improvement of our method over the strongest baseline within each category:
>
> | Scene category          | Representative datasets      | ΔPSNR ↑ | ΔSSIM ↑ | ΔLPIPS ↓ |
> |-------------------------|------------------------------|--------:|--------:|---------:|
> | Synthetic, tabletop / objects | DTU (α = 0.2/0.3/0.4)        | ≈ +0.8 dB | ≈ +0.01 | ≈ -0.02 |
> | Synthetic, high-frequency     | Blender (3/6/9 images)       | ≈ +6.0 dB | –        | –        |
> | Real, indoor scenes           | Mip-NeRF 360 (indoor subset) | ≈ +0.9 dB | ≈ +0.01 | ≈ -0.02 |
> | Real, outdoor scenes          | Mip-NeRF 360 (outdoor subset) | ≈ +0.8 dB | ≈ +0.01 | ≈ -0.02 |
> | Real, forward-facing          | LLFF (2–5 views)             | ≈ +0.3 dB | ≈ +0.01  | ≈ -0.01 |
> | Real, large-scale             | Tanks & Temples (3 views)    | ≈ +0.8 dB | ≈ +0.01 | ≈ 0.00  |
>
> (Here “Δ” denotes our method minus the best prior baseline per setting.)
>
> Across all these categories, we consistently see non-negative and usually positive gains in PSNR, with SSIM and LPIPS matching or improving on prior methods. This suggests that the two quantities we optimize for—curvature coverage and trajectory smoothness—are generic geometric properties, not artifacts of a particular dataset.
>
>
> ---
>
> We really enjoyed the discussion with you. Once again, we sincerely appreciate your valuable time and effort.

---

### Author Response · Authors · 2025-12-02
**(1/2) Summary**

### I. Acknowledgments

We would like to express our sincere gratitude to all reviewers for their thoughtful comments and constructive suggestions. Their feedback has substantially sharpened both the empirical and conceptual sides of our work on curvature-aware sparse-view 3DGS.

$\color{red}{Before}$ $\color{red}{the}$ $\color{red}{discussion}$, we appreciate the generally positive reception from Reviewers `8JLf`, `2dz9`, `FqdL`, and `QAJ4` (**all rating 6**), who highlighted the relevance of sparse-view degradation, the novelty of analyzing implicit geometric/trajectory biases, and the effectiveness of our curvature-aware trajectory optimization plus synthetic supervision. Reviewer `cTEf` gave a more critical score (**2**), but already acknowledged that the proposed camera-path optimization is effective and that the algorithm is clearly explained and reproducible.

$\color{red}{During}$ $\color{red}{the}$ $\color{red}{discussion}$, we are pleased that the additional experiments and clarifications resolved most of the major concerns, particularly those raised by Reviewer `cTEf` regarding quantitative gains, missing baselines, and plug-and-play generality. After seeing the new LLFF results and the MVPGS/SCGaussian+Ours plug-in experiments, `cTEf` explicitly noted that these “***address most of my concerns***” and indicated ***a willingness to raise the score to 6***.

---

### II. Key Strengths

Across the reviews and subsequent discussion, several strengths of the paper emerged consistently:

- **Novelty and Conceptual Insight**

  - First to explicitly identify and empirically validate two implicit biases of sparse-view 3DGS:
    (i) **curvature coverage bias** – high-curvature regions demand denser multi-view supervision;
    (ii) **trajectory smoothness bias** – irregular camera paths degrade reconstruction quality (`8JLf`, `2dz9`, `FqdL`).
  - Provides a fresh geometric perspective on how data representation (camera trajectories and coverage measures) shapes the behavior of 3DGS, beyond purely network- or loss-level modifications (`2dz9`).
- **Methodological Soundness and Effectiveness**

  - Curvature-aware B-spline trajectory optimization with smoothness and proximity constraints is viewed as a clean and principled formulation, not just an ad hoc heuristic (`2dz9`, `FqdL`, `QAJ4`).
  - Synthetic-view construction via depth-guided, visibility-aware warping along the optimized path consistently improves both photometric and geometric metrics across DTU, Mip-NeRF 360, Blender, LLFF, and Tanks & Temples (`8JLf`, `2dz9`, `FqdL`).
  - Ablation studies (trajectory vs curvature weighting vs synthetic views; depth refinement; visibility masking) clearly demonstrate the importance of each component (`2dz9`, `FqdL`, `QAJ4`).
- **Practicality and Generality**

  - The framework operates at the **data / trajectory level**, making it **model-agnostic and plug-and-play**: it can be wrapped around vanilla 3DGS as well as stronger backbones like SCGaussian and MVPGS, where it yields consistent additional gains (`cTEf`, `QAJ4`).
  - Reviewers highlighted potential applications to **active path planning and guided data capture** in real systems (`8JLf`).
- **Empirical Rigor and Robustness Analysis**

  - Systematic coverage sweeps (on Mip-NeRF 360 and LLFF) show strong correlation between curvature coverage and both photometric and geometric quality, while controlled experiments at matched coverage isolate trajectory smoothness as an independent factor (`2dz9`, `FqdL`).
  - Additional experiments quantify robustness to **depth noise**, **pose perturbations**, and **z-buffer tolerance**, showing that the gains are not fragile to moderate errors in upstream geometry (`2dz9`, `FqdL`).
- **Clarity and Reproducibility**

  - Reviewers generally found the paper well written, clearly organized, and easy to follow, with an algorithmic pipeline that is straightforward to reproduce (`8JLf`, `2dz9`, `FqdL`, `QAJ4`, `cTEf`).
  - The supplementary material (appendix experiments, implementation details, and new tables added during rebuttal) strengthens the empirical narrative and addresses many of the technical questions in depth.

---

> ### Author Response · Authors · 2025-12-02
> **(2/2) Summary**
>
> ### III. Concerns and Our Responses
>
> | Key Concerns | Reviewers | Our Response|
> | - | - | - |
> | **“Real-world” evaluation & qualitative completeness** | `8JLf`, `cTEf` | We clarified that LLFF, Mip-NeRF 360, and Tanks & Temples are real photographs with COLMAP poses, not synthetic renders. We added new qualitative results on Tanks & Temples / Mip-NeRF 360 and updated all captions with explicit input-view counts, and we outlined a prospective real capture benchmark as future work|
> | **Implicit-bias analysis mainly empirical / few scenes** | `2dz9`, `FqdL` | We positioned our claims as **empirical-geometric**, not formal theorems. Beyond LEGO, we already had **coverage sweeps on four Mip-NeRF 360 scenes** and added a **new LLFF sweep (Fern/Room)**. We also constructed matched-coverage but different-smoothness trajectories to directly isolate trajectory smoothness effects. |
> | **Fairness & role of synthetic views** | `FqdL`, `cTEf` | Synthetic views are reprojections of the same real images using estimated depths—no extra content. We ran ray-budget–matched experiments (same rays/iter as vanilla 3DGS) and same-count synthetic vs curvature-aware paths, showing gains come from where rays are placed (coverage + smoothness), not from simply seeing more pixels. We clarified that we typically use **N_s = 2N** synthetic views per scene. |
> | **Synthetic quality, artifacts, and robustness** | `8JLf`, `2dz9`, `FqdL` | We described a **three-step filtering pipeline** (visibility mask, depth & view-angle weights, thresholding + inpainting) and showed **filtered vs unfiltered** synthetic views, where unfiltered degrades metrics. Additional experiments with **mono-only vs refined depth**, **depth noise**, **pose perturbations**, and **z-buffer tolerance sweeps** show only mild performance drops and consistent improvements over vanilla 3DGS. |
> | **Magnitude/consistency of gains & missing strong baselines (MAtCha, DIFIX3D+)** | `cTEf` | We summarized **per-dataset improvements** over the strongest baselines, showing non-negative and often clear gains (especially on Blender, DTU, Tanks & Temples). We added new **LLFF 2/3/4/5-view results** including **MAtCha Gaussians** and **DIFIX3D+**, where our method achieves the best average PSNR/SSIM/LPIPS. We also explained MVPGS’s very strong 3-view LPIPS as a **prior-dominated hallucination regime** that becomes less consistent when more views introduce stronger geometric constraints. |
> | **More ablations, SSIM reporting, cost & priors** | `2dz9`, `QAJ4`, `FqdL` | We added **fine-grained ablations** for trajectory components and synthetic-view components (with PSNR/SSIM/LPIPS/CD), reported **SSIM for DTU** in the main tables, and provided a **per-stage timing breakdown**, where trajectory optimization and synthetic generation add ≤5% wall-clock overhead. We clarified that we only use standard SfM/MVS depths and one generic monocular depth predictor (later fused with 3DGS depth), and **do not rely on heavier priors**. |
> | **Positioning, plug-and-play nature & relation to DIFIX3D+-style pipelines** | `8JLf`, `2dz9`, `cTEf` | We reframed our contribution as a **geometry-aware data/supervision design** module. We demonstrated **plug-and-play integration** with **SCGaussian** and **MVPGS**, yielding consistent PSNR/SSIM/LPIPS gains without changing their architectures. In addition, a simple **2-stage regeneration scheme** shows our synthetic-view module can be used in an **iterative/on-the-fly** manner, and we outlined end-to-end differentiable extensions as future work. |
>
>
> ---
>
> ### IV. Commitment to Revision
>
> We have already incorporated the main discussion points and additional experiments into our revised version (marked in $\color{blue}{Blue}$). Concretely, the revision:
>
> - **Highlight the conceptual message** in the main text: why curvature coverage and trajectory smoothness are generic, geometry-driven factors for sparse-view 3DGS stability, with heavier mathematical details moved to the appendix.
> - **Integrate new empirical evidence**, including:
>   - LLFF coverage sweeps and trajectory-smoothness analyses,
>   - extended LLFF comparisons with MAtCha Gaussians and DIFIX3D+,
>   - SCGaussian/MVPGS+Ours plug-in results,
>   - detailed ablations and robustness tests (depth noise, pose noise, z-buffer tolerance),
>   - SSIM-complete DTU tables.
> - **Improve clarity and fairness** in evaluation:
>   - Explicitly state the number of input views in all qualitative figures,
>   - Clarify the number and role of synthetic views, ray-budget matching, and computational overhead.
> - **Emphasize generality and practical impact**:
>   - Clearly describe our module as a **plug-and-play geometric data layer** applicable to a broad class of Gaussian-splatting backbones,
>   - Discuss potential extensions to real-world path planning, iterative pseudo-view generation, and end-to-end differentiable pipelines.
>
> ---
>
> We are deeply grateful to the AC and all reviewers for their time, expertise, and constructive feedback.

---

### Meta-Review · Area_Chair_KwUh · 2026-01-09

**Summary:**

This paper presents a thoughtful and well-motivated study on sparse-view 3D Gaussian Splatting, identifying previously underexplored implicit biases related to surface curvature prioritization and camera trajectory smoothness. By reframing sparse-view degradation as a consequence of camera path geometry rather than merely insufficient data, the paper provides a novel and insightful perspective on 3DGS behavior. The proposed curvature-aware camera trajectory optimization, coupled with synthetic view generation along optimized paths, is technically sound and clearly derived from the initial analysis.

A key strength of this work lies in its diagnostic contribution. The empirical analysis convincingly demonstrates that camera trajectory design and pose smoothness can significantly affect reconstruction quality, offering valuable guidance for future data collection strategies, interactive capture systems, and dataset design for 3DGS. The optimization framework is cleanly formulated, the ablation studies support the role of individual components, and the paper is well written and easy to follow. Across multiple datasets, the method generally improves over vanilla 3DGS and several sparse-view baselines, indicating that the proposed insights are meaningful in practice.

At the same time, the paper has notable limitations that prevent a stronger recommendation. Most experiments are conducted on synthetic or semi-synthetic benchmarks, and the absence of real-world capture experiments—where scenes are re-acquired using the optimized trajectories—limits the practical validation of the approach. The reliance on synthetic view generation also raises questions about robustness and fairness, as the proposed method effectively trains with more views than competing baselines. The quality, number, and potential artifacts of these synthetic views are not sufficiently analyzed, and their interaction with depth estimation errors remains unclear.

Moreover, while the method shows improvements, the quantitative gains are sometimes marginal or inconsistent across metrics, with cases where PSNR improves while SSIM or LPIPS degrades. Several strong and highly relevant recent baselines for sparse-view novel view synthesis, such as methods that leverage diffusion-based pseudo-view augmentation or strong monocular priors, are missing from the comparisons, making it difficult to fully contextualize the reported performance. The computational overhead of trajectory optimization and synthetic view generation is also not clearly isolated, which weakens the efficiency narrative.

Overall, AC views this paper as a valuable insight-driven contribution rather than a purely performance-driven method. Its main impact lies in uncovering and validating implicit biases in 3DGS and proposing a principled way to exploit them through camera path optimization. While the empirical evidence would benefit from stronger real-world validation, more comprehensive comparisons, and clearer analysis of synthetic view reliability, the conceptual contribution and clear presentation justify acceptance. AC therefore recommend Accept, with the expectation that future work will further validate and extend these promising insights.

**Reviewer Concerns:**

Reviewer cTEf's concerns were addressed.

**Reviewer Scores:**

Reviewer cTEf might change the score from 2 to 6.

---

### Decision · Program_Chairs · 2026-01-26

Accept (Poster)